# A pair of effectors encoded on a conditionally dispensable chromosome of *Fusarium oxysporum* suppress host-specific immunity

Yu Ayukawa [1,2,7], Shuta Asai [1,3,7 ✉], Pamela Gan[1], Ayako Tsushima [1,5], Yasunori Ichihashi[1,6], Arisa Shibata[1], Ken Komatsu[2], Petra M. Houterman[4], Martijn Rep [4], Ken Shirasu [1] & Tsutomu Arie [2 ✉]

Many plant pathogenic fungi contain conditionally dispensable (CD) chromosomes that are associated with virulence, but not growth in vitro. Virulence-associated CD chromosomes carry genes encoding effectors and/or host-specific toxin biosynthesis enzymes that may contribute to determining host specificity. *Fusarium oxysporum* causes devastating diseases of more than 100 plant species. Among a large number of host-specific forms, *F. oxysporum* f. sp. *conglutinans* (*Focn*) can infect Brassicaceae plants including Arabidopsis (*Arabidopsis thaliana*) and cabbage. Here we show that *Focn* has multiple CD chromosomes. We identified specific CD chromosomes that are required for virulence on Arabidopsis, cabbage, or both, and describe a pair of effectors encoded on one of the CD chromosomes that is required for suppression of Arabidopsis-specific phytoalexin-based immunity. The effector pair is highly conserved in *F. oxysporum* isolates capable of infecting Arabidopsis, but not of other plants. This study provides insight into how host specificity of *F. oxysporum* may be determined by a pair of effector genes on a transmissible CD chromosome.

[1] Center for Sustainable Resource Science, RIKEN, Yokohama, Kanagawa, Japan. [2] Laboratory of Plant Pathology, Graduate School of Agriculture, Tokyo University of Agriculture and Technology (TUAT), Fuchu, Tokyo, Japan. [3] PRESTO, Japan Science and Technology Agency, Kawaguchi, Saitama, Japan. [4] Molecular Plant Pathology, Swammerdam Institute for Life Sciences, University of Amsterdam, Amsterdam, The Netherlands. [5] Present address: John Innes Centre, Norwich, UK. [6] Present address: RIKEN BioResource Research Center, Tsukuba, Ibaraki, Japan. [7] These authors contributed equally: Yu Ayukawa, Shuta Asai. ✉email: shuta.asai@riken.jp; arie@cc.tuat.ac.jp

Pathogenic fungi often carry chromosomes that are not necessary for growth in the non-pathogenic state[1,2]. Analogous to the well-characterized virulence plasmids in bacteria, the number of these dispensable chromosomes in individual isolates can vary. In plant pathogenic fungi, dispensable chromosomes that are associated with virulence are generally referred to as supernumerary, 'B', or conditionally dispensable (CD) chromosomes[1]. When pathogenic fungi lack CD chromosomes, they can grow in vitro, but often exhibit attenuated or no virulence[1,2]. The functions of CD chromosomes in some plant pathogenic fungi are associated with suppression or deactivation of host-specific factors. In *Fusarium solani*, for example, a CD chromosome carries phytoalexin detoxifying genes[3]. In contrast, the CD chromosomes of *Alternaria alternata* and *Cochliobolus carbonum* harbor host-specific toxin genes[2,4]. Therefore, CD chromosomes can be crucial determinants of host specificity that are defined by phytotoxin activity or by defense against chemicals such as phytoalexins.

*Fusarium oxysporum* causes devastating diseases of more than 100 plant species, including economically important crops such as tomato, banana, and melon[5]. Individual isolates of *F. oxysporum* have different host ranges and are classified into *formae speciales* (ff. spp.) based on the susceptibility of plant species to infection. Although much is known about the genetics and pathology of *F. oxysporum*, the precise molecular mechanisms of host specificity remain unclear. So far, CD chromosomes have been identified in the tomato-infecting pathogen *F. oxysporum* f. sp. *lycopersici* (*Fol*) and in *F. oxysporum* f. sp. *radicis-cucumerinum* (*Forc*), a cucurbit-infecting pathogen[6–8]. The *Fol* isolate 4287 and the *Forc* isolate 016 each contain a single virulence-associated CD chromosome that is transferable to other isolates[7–9]. Horizontal transfer of the CD chromosomes from *Fol*4287 or *Forc*016 converts non-pathogenic *F. oxysporum* isolates into pathogens of their respective hosts[6,7,9]. Part of this phytopathogenic conversion is often due to the expression of CD-encoded effectors that modulate host immunity against infection, such as Secreted In Xylem (SIX) effectors that are, as their name indicates, secreted into xylem elements during infection[10,11]. A total of fourteen *SIX* genes (*SIX1* to *14*) have been identified from *Fol*[10]. The CD chromosome of *Fol*4287 contains all of the *SIX* genes except *SIX4*, which is not present in *Fol*4287[6,12], but is present in certain other *Fol* isolates. The CD chromosome of *Forc*016 contains *SIX6*, *SIX9*, *SIX11*, and *SIX13* homologs[7]. *SIX1*, *SIX3*, *SIX5*, and *SIX6* from

*Fol* are involved in overcoming resistance in tomato and the *SIX6* homolog from *Forc*016 is crucial for virulence in cucumber[7,10]. However, their molecular mechanisms as virulence factors are as yet unknown.

Arabidopsis-infecting isolates of *F. oxysporum* are useful as a model pathosystem. There are at least three ff. spp. that cause disease on Arabidopsis: f. sp. *conglutinans*, f. sp. *matthiolae*, and f. sp. *raphani*[13]. *F. oxysporum* f. sp. *conglutinans* (*Focn*) can also infect other Brassicaeae plants such as cabbage (*Brassica oleracea* var. *capitata*). The *SIX1* gene is required for full virulence on cabbage in *Focn*[14], but the *Focn* factor(s) that are required for virulence on Arabidopsis have not been identified. We have previously shown that the *Focn* isolate Cong:1-1 (*Focn*Cong:1-1) harbors *SIX1*, *SIX4*, *SIX8*, and *SIX9* homologs on multiple chromosomes of different sizes[15]. Although these chromosomes are presumed to be conditionally dispensable in *Focn*, their status as CD chromosomes has not been established.

Here we report, through analyses of chromosome-deficient *Focn*Cong:1-1 strains and through horizontal chromosome transfer, that *Focn*Cong:1-1 has multiple CD chromosomes. Importantly, we identified individual CD chromosomes that are required for virulence on Arabidopsis, cabbage, or both. Furthermore, we identified a pair of effector genes on a CD chromosome that are required for suppression of Arabidopsis-specific phytoalexin-based immunity.

## Results

**Chromosome-level genome assembly of *Focn*Cong:1-1.** We assembled the *Focn*Cong:1-1 genome sequence into 198 contigs with an N50 of 1.271 Mb. To improve contiguity, we further performed optical mapping using two restriction enzymes. The final assembly consisted of 22 scaffolds (SCs) with an N50 SC length of 4865 kb and a 99.1% complete BUSCO score (Table 1). For gene prediction, we generated transcriptome data from axenic culture and plant infections, resulting in a total of 21,781 genes, among which are eight presumptive effector genes (*SIX1*, *SIX4*, *SIX8*, *SIX9*, and *FOA1-FOA4*) that were previously known from Arabidopsis-infecting *F. oxysporum*[16], as well as the homologous genes of *FOA1* and *FOA4*, which were named *FOA1b* and *FOA4b*, respectively. We did not detect homologs of any other *SIX* genes. To find unknown effectors, 1467 putative secreted proteins were screened for proteins with an effector-like structure using the EffectorP v1 and/or v2 algorithm[17,18]. A total of 263 secreted proteins were predicted as effectors by both EffectorP v1 and v2. This prediction did not include FOA1, which is involved in the suppression of pattern-triggered immunity[16], nor its homolog FOA1b. Therefore, a total of 265 proteins, including FOA1 and FOA1b, were defined as high-confidence effector candidates (Table 1 and Supplementary Data 1).

The *F. oxysporum* genome is composed of core genomic regions that are conserved among *Fusarium* species, and additional accessory genomic regions that are conserved in certain isolates[19]. Comparative analysis with the *Fol*4287 genome as a reference indicated that (i) the *Focn*Cong:1-1 SCs have no homology with known accessory genomic regions in *Fol*4287 (chr01B; chr02B; chr03; chr06; chr14; chr15)[6], (ii) similarly, there are genomic regions of *Focn*Cong:1-1 that have no homology with *Fol*4287, and (iii) the non-homologous genomic regions are enriched in transposable elements (TEs) (Fig. 1a). All known effector genes except *FOA4* are located in the TE-rich genomic region in *Focn*Cong:1-1 as follows: *SIX1* (in SC8), *SIX4* (SC9), *SIX8* (SC10), *SIX9* (SC3), *FOA1* (SC5), *FOA1b* (SC10), *FOA2* (SC9), *FOA3* (SC3), and *FOA4b* (SC10) (Supplementary Fig. 1). *FOA4* (SC12) may be a pseudogene since it is not expressed either in vitro or in planta (Supplementary Data 1 and 2). TEs are

**Table 1 *Focn*Cong:1-1 genome statistics.**

| Statistics | Value |
| --- | --- |
| Assembly stats | |
| Assembly size (Mb) | 68.8 (72.2)[a] |
| No. of scaffolds | 22 |
| No. of non-scaffolded contigs | 125 |
| Max scaffold length (kb) | 7,006 |
| N50 scaffold length (kb) | 4,865 |
| GC content (%) | 48.40 |
| BUSCO coverage (%) | 99.1 |
| Gene models | |
| Total no. of genes | 21,781 |
| Total no. of proteins | 22,094 |
| No. of genes encoding secreted proteins | 1,467 |
| No. of high-confidence effector candidates: | 265[b,c] |
| Predicted by both EffectorP v1 and v2 | 263 |
| Predicted by either EffectorP v1 or v2 | 393 |

[a]The number in parentheses is total assembly size including non-scaffolded contigs.
[b]FOA1 and FOA1b are added to the list of effector candidates predicted by both EffectorP v1 and v2.
[c]Note that some genes encode identical amino acid.

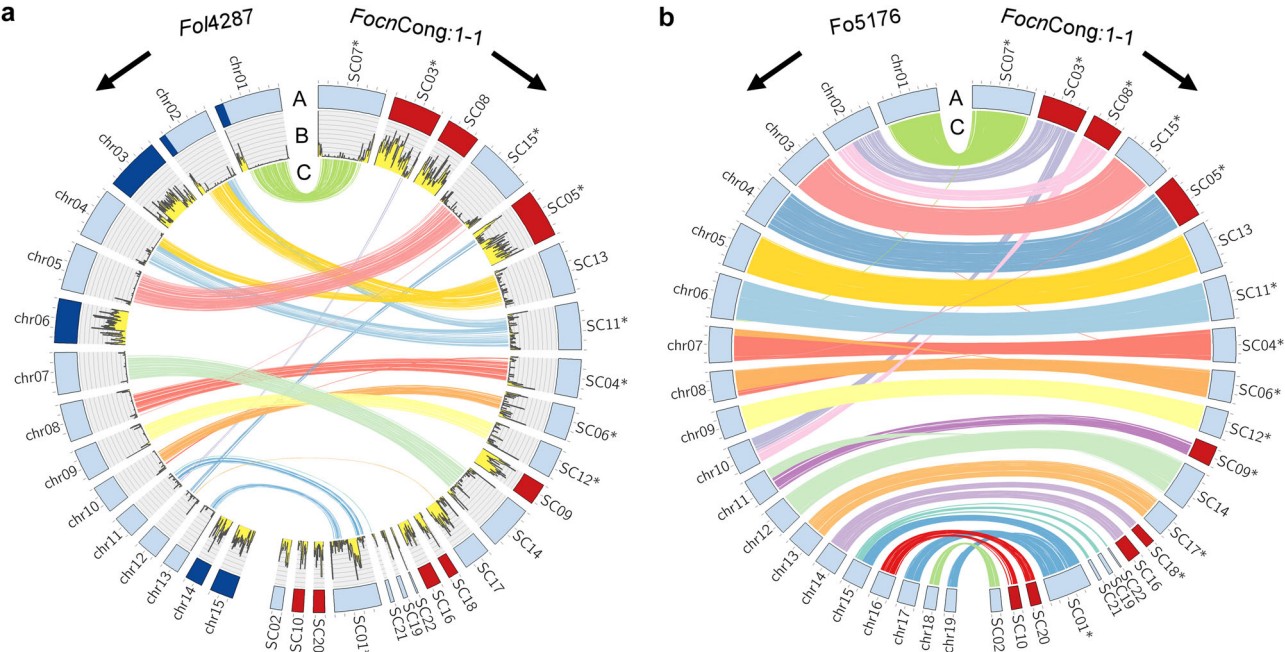

**Fig. 1 Comparison of whole genome assemblies among *Focn*Cong:1-1, *Fol*4287, and Fo5176.** Whole genome assemblies were compared between *Fol*4287 and *Focn*Cong:1-1 (**a**) and between Fo5176 and *Focn*Cong:1-1 (**b**). Ring A: Circular representation of the pseudomolecules. Red and dark blue indicate dispensable genomic regions in *Focn*Cong:1-1 and known accessory regions in *Fol*4287 (chr1B; chr2B; chr3; chr6; chr14; chr15)[6], respectively. Light blue indicates the remaining regions. Ring B: Distribution of transposable elements (TEs) in 50 kb windows. Proportions of sequences of respective *Focn*Cong:1-1 SCs associated with TEs are shown in Supplementary Fig. 1. Ring C: Syntenic regions (>95% identity, 30 kb) between *Fol*4287 and *Focn*Cong:1-1 assemblies (**a**) and between Fo5176 and *Focn*Cong:1-1 assemblies (**b**). Asterisks indicate reverse-complemented scaffolds (SCs) for visual clarity. Ticks on bands represent 1 Mb.

suspected to be involved in the generation of genomic variations leading to environmental adaptation and, in the case of pathogens, they may have been involved in the acquisition of the ability to infect particular hosts[20]. Therefore, chromosomes containing TE-enriched genomic regions have a high potential to be CD chromosomes.

Recently, a chromosome-level genome assembly of the Arabidopsis-infecting *F. oxysporum* isolate Fo5176 was reported[21]. The genomes of *Focn*Cong:1-1 and Fo5176 are very similar, sharing from 93.2% to 94.3% of their total scaffold/contig lengths (>95% identity, 10 kb). Synteny analysis revealed that (i) SC16 and SC18 of *Focn*Cong:1-1 correspond to chromosome 14 (chr14) of Fo5176, and (ii) SC10 and SC20 to chr16 (Fig. 1b), indicating that these SCs constitute, or contribute to the respective chromosomes. Due to the observations (i) and (ii) above, we refer to the chromosomes carrying these sequences as chr$^{SC16/SC18}$ and chr$^{SC10/SC20}$ in *Focn*Cong:1-1, respectively.

**FocnCong:1-1 has multiple CD chromosomes**. To identify CD chromosomes of *Focn*Cong:1-1, we generated chromosome-deficient strains by treatment with a mitosis inhibitor benomyl[7,22]. For the parental strain, we utilized the previously generated strain *Focn*Cong:1-1 ΔSIX4, in which *SIX4*, located in SC9, had been replaced with a hygromycin B resistance gene (*hph*) cassette[23]. After benomyl treatment, we obtained six hygromycin B-sensitive mutants (HS1 to HS6; Supplementary Figs. 2 and 3). To confirm the loss of dispensable genomic regions, we sequenced the genomes of *Focn*Cong:1-1 ΔSIX4 and each of the HS mutants. As expected, *Focn*Cong:1-1 ΔSIX4 maintained SC9 carrying *hph*, but all of the HS mutants had lost SC9 (Fig. 2a, b). We also found that (i) SC3 was absent in HS2, HS3, and HS4, (ii) SC5 and SC8 were lost only in HS6, (iii) chr$^{SC10/SC20}$ was missing in HS3, HS4, HS5, and HS6, and (iv)

chr$^{SC16/SC18}$ was lost only in HS4. In addition, duplication of SC2 and part of SC2 and SC17 occurred in HS1, HS5, and HS2, respectively (Fig. 2a).

Among the *Focn*Cong:1-1 HS mutants, there was no appreciable difference in colony size, but there was a significant difference in conidial formation (Fig. 2c, and Supplementary Figs. 2c and 4). *Focn*Cong:1-1 HS1 (ΔSC9) showed no difference in conidial formation or virulence on either Arabidopsis or cabbage compared to the parent strain ΔSIX4, indicating that SC9 is involved in neither conidial formation nor virulence (Fig. 2c, d and Supplementary Fig. 5). *Focn*Cong:1-1 mutants without SC3 (HS2, HS3, and HS4) had attenuated virulence on Arabidopsis and cabbage, but also had reduced ability to form conidia (Fig. 2c, d and Supplementary Fig. 5), suggesting that SC3 positively regulates conidial formation. To our surprise, loss of chr$^{SC10/SC20}$ in *Focn*Cong:1-1 HS5 and HS6 increased conidial formation (Fig. 2c), but reduced virulence on Arabidopsis (Fig. 2d). Interestingly, *Focn*Cong:1-1 HS6 (ΔSC5/SC8/SC9/chr$^{SC10/SC20}$) lost virulence on both Arabidopsis and cabbage, whereas HS5 (ΔSC9/chr$^{SC10/SC20}$) retained virulence on cabbage, but not on Arabidopsis (Fig. 2d). These data indicate that chr$^{SC10/SC20}$ is required for disease progression on Arabidopsis, but that SC5 and/or SC8 are involved only in causing disease in cabbage. Therefore, SC3, SC5, SC8, and chr$^{SC10/SC20}$ are CD chromosomes affecting disease levels, with SC3 and chr$^{SC10/SC20}$ being also associated with conidial formation.

**FocnCong:1-1 CD chromosomes are transferable**. We investigated whether *Focn*Cong:1-1 CD chromosomes are transferable under laboratory conditions, and what their effect might be on virulence. *Focn*Cong:1-1 HS6 lost multiple virulence-associated CD chromosomes (SC5, SC8, and chr$^{SC10/SC20}$) along with virulence on both Arabidopsis and cabbage (Fig. 2). Strain HS6

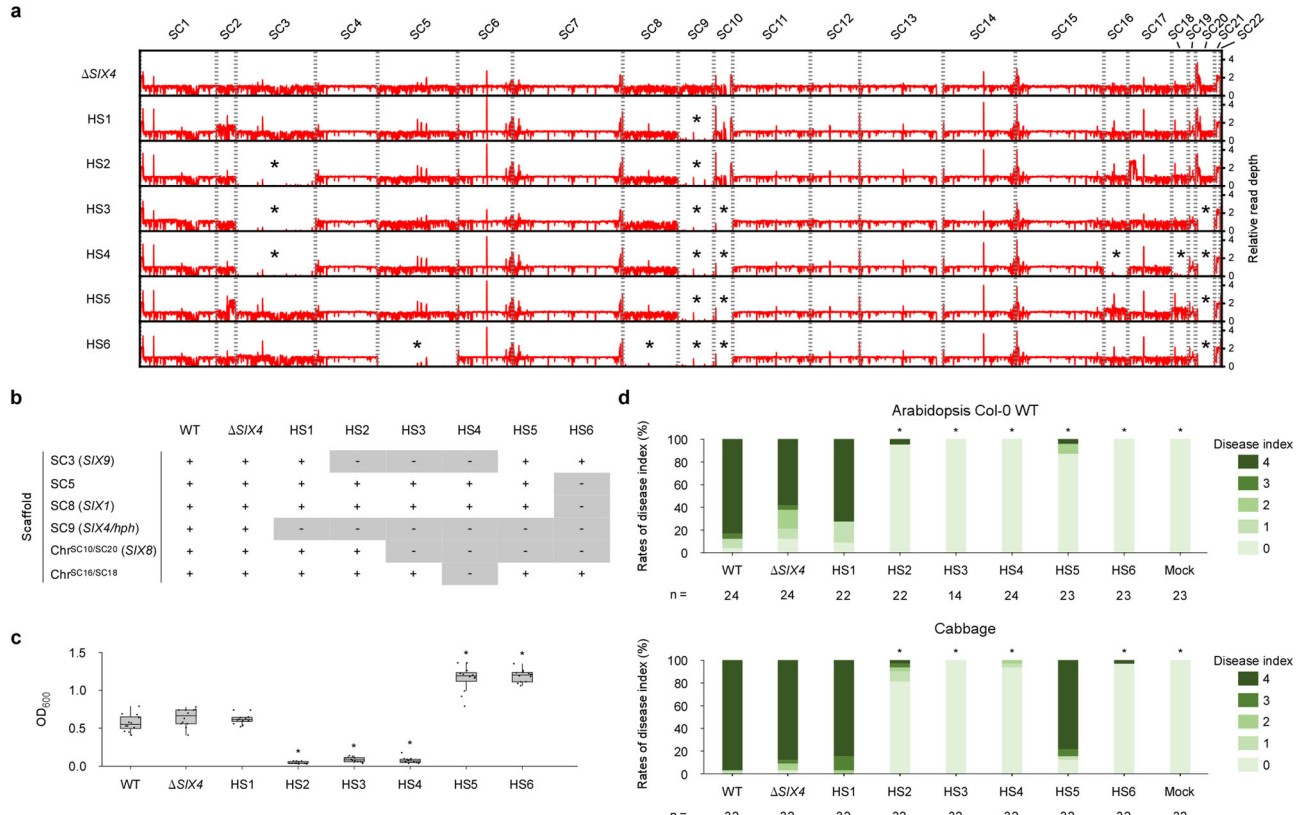

**Fig. 2 Effects of loss of conditionally dispensable chromosomes on conidial formation and virulence in *Focn*Cong:1-1. a** Relative read mapping depths of *Focn*Cong:1-1 Δ*SIX4* and HSs whole genomes. Asterisks indicate scaffold (SC)-level deletion. **b** Loss patterns of SC in *Focn*Cong:1-1 HSs. + and – represent maintained- and lost-SCs, respectively. *SIX*s located on particular SCs are shown in parentheses. **c** Conidial formation in *Focn*Cong:1-1 WT, Δ*SIX4* and HSs. $OD_{600}$ of conidial suspensions was measured from six colonies after 17 days of incubation on potato dextrose agar. Results of two independent experiments were combined and a total of twelve biological replicates per isolate are plotted. Boxplots indicate median value, estimated 25th and 75th percentiles, and whiskers represent 1.5 times the interquartile range. Asterisks represent significant differences from *Focn*Cong:1-1 Δ*SIX4* (*$p < 0.0001$, Welch's t-test). **d** Virulence of *Focn*Cong:1-1 WT, Δ*SIX4* and HSs to Arabidopsis and cabbage. Disease index was scored as described in Methods. Results of at least two experiments were combined. *n* denotes the number of plants investigated. Asterisks represent significant differences from *Focn*Cong:1-1 Δ*SIX4* (*$p < 0.001$, Mann–Whitney U-test). Representative images of Arabidopsis and cabbage at 28 dpi are shown in Supplementary Fig. 5.

could therefore be used to determine the effects of chromosome transfer on virulence. A phleomycin-resistant *Focn*Cong:1-1 HS6 strain (HS6-BLE) was generated by introducing the phleomycin resistance gene (*ble*) (Supplementary Figs. 6a and 7). We co-incubated *Focn*Cong:1-1 HS6-BLE with hygromycin B-resistant *Focn*Cong:1-1 Δ*SIX4* as a donor and selected four colonies (HCT1 to HCT4) that were resistant to both phleomycin and hygromycin B. There was no apparent difference in morphology or colony size (i.e., growth rate) in any of the presumptive *Focn*Cong:1-1 recipients (HCT1 to HCT4; Supplementary Fig. 6b). We confirmed chromosome transfer by contour-clamped homogeneous electric field (CHEF) electrophoretic karyotyping as well as PCR (Fig. 3a and Supplementary Figs. 6–8). *SIX1* (in SC8) and *hph* (in SC9) were detected in all *Focn*Cong:1-1 recipients, whereas *SIX8* (in SC10) and an SC20 marker (*FocnCong_v011766*) were detected only in HCT1 (Supplementary Fig. 6a). We did not detect the SC5 marker *Focn-Cong_v016149* in any recipient (Supplementary Fig. 6a). These results indicate that at least SC8, SC9, and chr^SC10/SC20 are transferable. Conidial formation of the three *Focn*Cong:1-1 recipients HCT2, HCT3, and HCT4, which acquired SC8 and SC9, was comparable to that of HS6-BLE. In contrast, *Focn*Cong:1-1 HCT1, which received chr^SC10/SC20, SC8, and SC9, produced significantly fewer conidia than HS6-BLE, a phenotype similar to the donor Δ*SIX4* (Fig. 3b). Because SC8 and SC9 are not involved in conidial formation (Fig. 2c), these results suggest a negative

involvement of chr^SC10/SC20 in conidial formation. All *Focn*Cong:1-1 recipients (i.e., HCT1 to HCT4) that acquired SC8 and SC9 also regained virulence against cabbage (Fig. 3c and Supplementary Fig. 9). Because SC9 is not involved in virulence (Fig. 2d), we conclude that SC8 is necessary and sufficient for virulence to cabbage. In the case of Arabidopsis, *Focn*Cong:1-1 HCT1 showed higher virulence than the other recipients (HCT2, HCT3, and HCT4; Fig. 3c and Supplementary Fig. 9). Taken together with the pathology results of the chromosome-deficient *Focn*Cong:1-1 mutants (HS1 to HS6; Fig. 2), it is likely that chr^SC10/SC20 is required for virulence on Arabidopsis.

**Chr^SC10/SC20 is involved in suppression of *CYP79B2/CYP79B3*-mediated immunity.** A CD chromosome from the Arabidopsis-infecting anthracnose fungus *Colletotrichum higginsianum* has been reported to be involved in suppression of plant immunity that is dependent on tryptophan (Trp)-derived secondary metabolites[24]. We investigated whether CD chromosomes of *Focn*Cong:1-1 encode products that also suppress specific immunity. For this experiment, we used the Arabidopsis double mutant *cyp79b2/cyp79b3* that lacks the ability to synthesize Trp-derived secondary metabolites[25]. Among the chromosome-deficient *Focn*Cong:1-1 mutants (HS2 to HS6) with attenuated virulence to Arabidopsis Col-0 WT, only *Focn*Cong:1-1 HS5 (ΔSC9/chr^SC10/SC20) showed the same level of virulence on

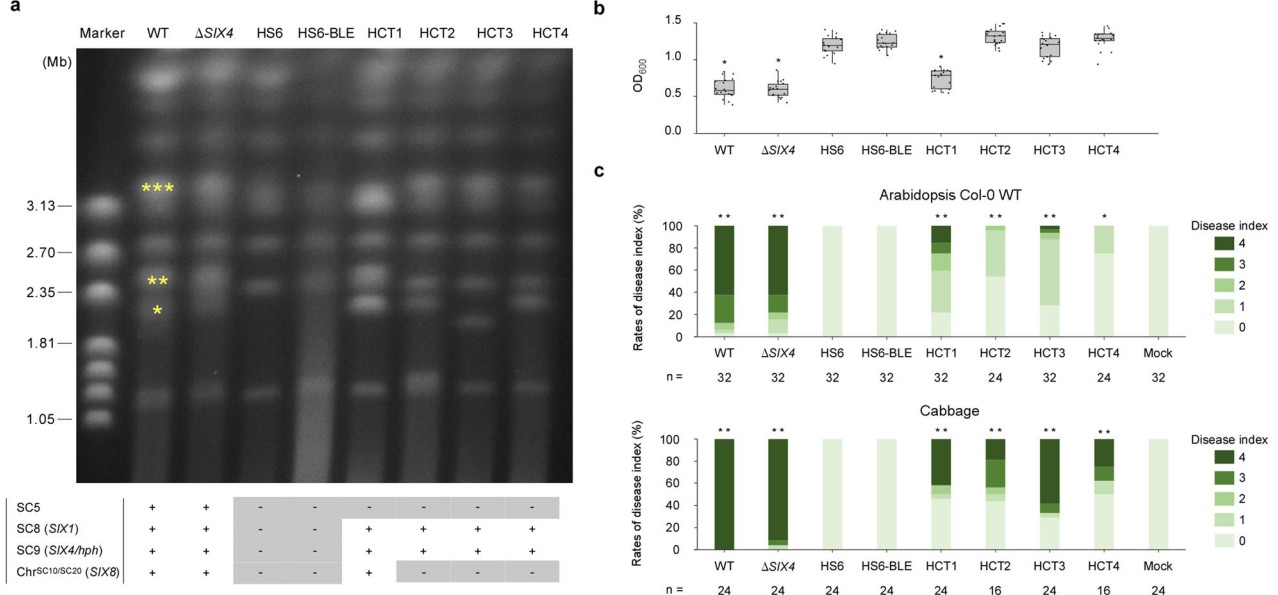

**Fig. 3 Effects of chromosome transfer on conidial formation and virulence in *Focn*Cong:1-1 HS6. a** Electrophoretic karyotype of *Focn*Cong:1-1 WT, ΔSIX4, HS6, HS6-BLE, and HCTs. Asterisks indicate chromosomes on which *SIX* genes are located[15]: *SIX4, **SIX8, ***SIX1. The table indicates the scaffold (SC) patterns confirmed by PCR as shown in Supplementary Fig. 6a. + and – represent maintained- and lost-SCs, respectively. *SIX*s located on SCs are shown in parentheses. **b** Conidial formation in *Focn*Cong:1-1 WT, ΔSIX4, HS6, HS6-BLE, and HCTs. $OD_{600}$ of conidial suspension was measured from six colonies after 17 days of incubation on potato dextrose agar. Results of three independent experiments were combined and a total of 18 biological replicates are plotted. Boxplots indicate median value, estimated 25th and 75th percentiles, and whiskers represent 1.5 times the interquartile range. Asterisks represent significant differences from *Focn*Cong:1-1 HS6-BLE (*$p < 0.0001$, Welch's t-test). **c** Virulence of *Focn*Cong:1-1 WT, ΔSIX4, HS6, HS6-BLE, and HCTs on Arabidopsis and cabbage. Disease index was scored as described in Methods. Results of at least two independent experiments were combined. *n* denotes the number of plants investigated. Asterisks represent significant difference from *Focn*Cong:1-1 HS6-BLE (**$p < 0.001$, *$p < 0.01$ Mann–Whitney U-test). Representative images of Arabidopsis and cabbage at 28 dpi are shown in Supplementary Fig. 9.

*cyp79b2/cyp79b3* plants as was observed for its parent strain ΔSIX4 (Fig. 4a and Supplementary Fig. 10a, b). These results suggest that chr^SC10/SC20^ plays a key role in suppressing Trp-derived secondary metabolite-dependent immunity. *Focn*Cong:1-1 HS6 (ΔSC5/SC8/SC9/chr^SC10/SC20^) was substantially less virulent on *cyp79b2/cyp79b3* plants. This is likely because SC5 or SC8 are involved in virulence other than through suppression of Trp-based immunity. In addition, we found that the *cyp79b2/cyp79b3* double mutant was resistant to all tested SC3-deficient *Focn*Cong:1-1 mutants (HS2 to HS4; Fig. 4a and Supplementary Fig. 10a, b), possibly due to some deficiency of conidial formation in these mutants (Fig. 2c).

To investigate which step or steps of infection the CD chromosomes contribute to, a histological analysis was performed using GFP-labeled *Focn*Cong:1-1 strains in Arabidopsis. *Focn*Cong:1-1 ΔSIX4-GFP always colonized xylem vessels of roots, often reaching stem elements in Arabidopsis WT, whereas *Focn*Cong:1-1 HS2-GFP lacking SC3 germinated on root surfaces but showed almost no colonization in xylem vessels of roots or stems (Fig. 4b), confirming its deficiency in growth in planta. *Focn*Cong:1-1 HS5-GFP (ΔSC9/chr^SC10/SC20^) colonized root xylem vessels, but the frequency of stem colonization was low in Arabidopsis WT. In *cyp79b2/cyp79b3* double mutants, however, *Focn*Cong:1-1 HS5-GFP frequently colonized the stems as was observed for *Focn*Cong:1-1 ΔSIX4 in WT (Fig. 4b). These results suggest that chr^SC10/SC20^ is implicated in the ability to colonize beyond root xylem vessels into stems, and conversely that CYP79B2/CYP79B3 participate in inhibition of *Focn*Cong:1-1 colonization of stems.

To determine whether CYP79B2/CYP79B3-based antibiotics such as isothiocyanate, camalexin, or 4-hydroxyindole-3-carbonyl nitrile (4-OH-ICN)[26,27] are associated with resistance, we conducted bioassays with *Focn*Cong:1-1 HS5 (ΔSC9/chr^SC10/SC20^) on three Arabidopsis mutants (*pen2*, *pad3*, and *cyp82c2*) that are unable to produce isothiocyanate, camalexin, and 4-OH-ICN, respectively[26,27] (Fig. 4c). The *pad3* mutant, but not *pen2* or *cyp82c2*, was more susceptible to *Focn*Cong:1-1 HS5 than Col-0 WT (Fig. 4d and Supplementary Fig. 10c), suggesting that camalexin, but not isothiocyanate or 4-OH-ICN, is involved in resistance to *Focn*Cong:1-1. Importantly, camalexin is produced in Arabidopsis, but not in cabbage[28]. Because *Focn*Cong:1-1 HS5 (ΔSC9/chr^SC10/SC20^) had attenuated virulence on Arabidopsis but full virulence on cabbage (Fig. 2d), chr^SC10/SC20^ is likely to contribute to suppression of Arabidopsis-specific immunity, specifically camalexin, to establish infection.

**A pair of effectors are involved in virulence on Arabidopsis.** Because chr^SC10/SC20^ is likely to encode effectors that contribute to suppression of Arabidopsis-specific immunity, we searched for genes encoding potential effectors, and found a total of twelve effector candidate genes located on chr^SC10/SC20^ (Supplementary Data 1). Expression profiling revealed that *FocnCong_v001893* (*SIX8*) and *FocnCong_v001894* were highly expressed during infection (Fig. 5a and Supplementary Data 1). Interestingly, *SIX8* is adjacent to *FocnCong_v001894*, with an intergenic distance of 1057 bp on SC10 (Fig. 5b). The intergenic region contains a miniature impala inverted repeat (mimp-IR) sequence, which is related to TE sequences (Fig. 5b and Supplementary Fig. 11). A mimp-IR is also often located in the upstream regions of *SIX* and other effector candidate genes in *Fol*, *Forc*, and the melon-infecting pathogen *F. oxysporum* f. sp. *melonis*[7,12,29]. To determine whether *SIX8* and *FocnCong_v001894* are involved in virulence on Arabidopsis, a genome fragment containing the *SIX8*-*FocnCong_v001894* locus was introduced into *Focn*Cong:1-1

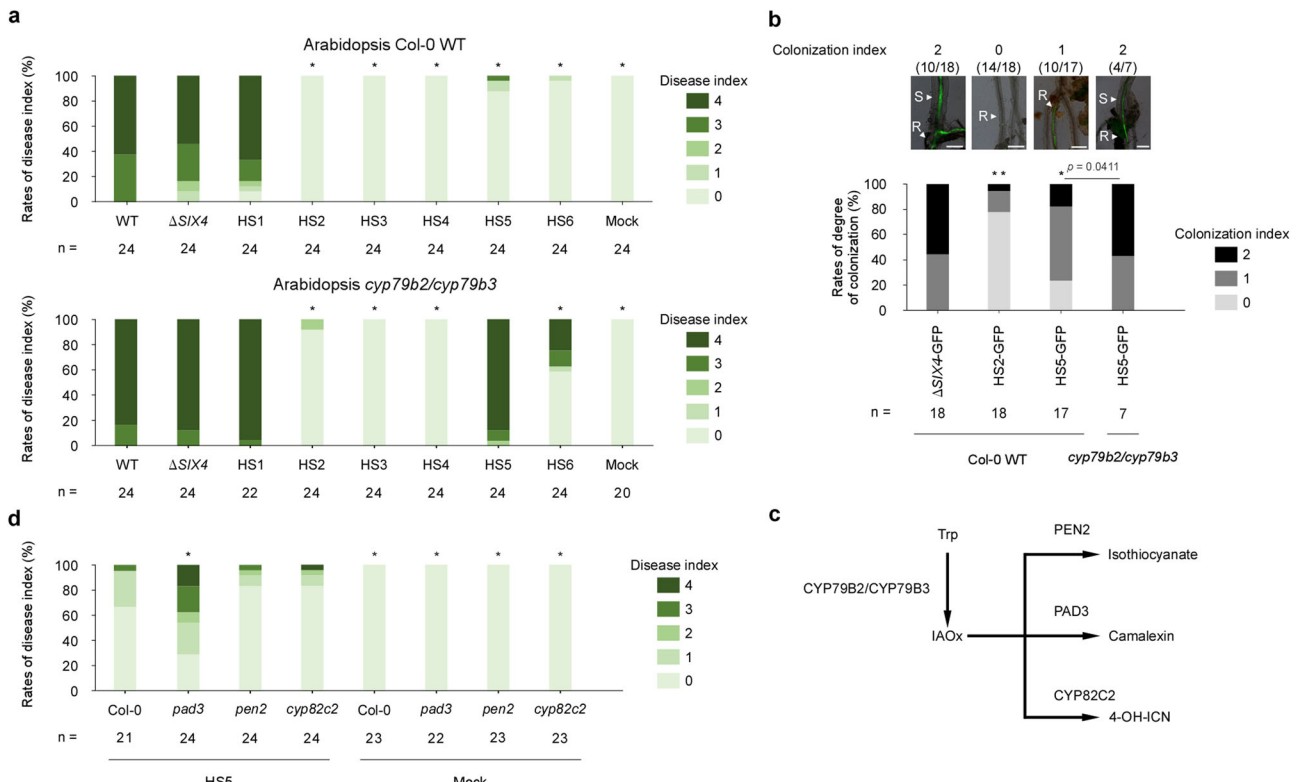

**Fig. 4 Involvement of chr$^{SC10/SC20}$ in suppression of *CYP79B2/CYP79B3*-mediated immunity. a** Virulence of *Focn*Cong:1-1 WT, Δ*SIX4* and HSs on Arabidopsis Col-0 WT and the *cyp79b2/cyp79b3* double mutant. Disease index was scored as described in Methods. Results of three independent experiments were combined. *n* denotes the number of plants investigated. Asterisks represent significant difference from *Focn*Cong:1-1 Δ*SIX4* (*$p < 0.001$, Mann–Whitney U-test). Representative images of Arabidopsis at 28 dpi are shown in Supplementary Fig. 10a, b. **b** Infection phenotypes of *Focn*Cong:1-1 Δ*SIX4*, HS2 and HS5 in Arabidopsis. Colonization index of Arabidopsis Col-0 WT and *cyp79b2/cyp79b3* double mutant inoculated with GFP-labeled *Focn*Cong:1-1 Δ*SIX4* (Δ*SIX4*-GFP), HS2 (HS2-GFP) or HS5 (HS5-GFP) at 12 dpi was scored from 0 to 2: 0, germination or colonization on root surface; 1, colonization in xylem vessels of roots, 2, transition from roots to stems. Results of at least two independent experiments were combined. *n* denotes the number of plants investigated. Asterisks represent significant differences from *Focn*Cong:1-1 Δ*SIX4*-GFP (**$p < 0.001$, *$p < 0.01$, Mann–Whitney U-test). Representative images of root or stem of Arabidopsis at 12 dpi are shown above each bar. Scale bars indicate 200 μm. S stems, R roots. **c** A simple scheme of biosynthesis of tryptophan (Trp)-derived defense compounds in Arabidopsis. IAOx indole-3-acetaldoxime, 4-OH-ICN 4-hydroxy-indole carbonyl nitrile. **d** Virulence of *Focn*Cong:1-1 HS5 on Arabidopsis Col-0 WT, *pad3*, *pen2*, and *cyp82c2* mutants. Disease index was scored as described in Methods. Results of three independent experiments were combined. *n* denotes the number of plants investigated. Asterisks represent significant difference from Arabidopsis Col-0 WT infected with *Focn*Cong:1-1 HS5 (*$p < 0.01$, Mann–Whitney U-test). Representative images of Arabidopsis at 28 dpi are shown in Supplementary Fig. 10c.

HS5 (ΔSC9/chr$^{SC10/SC20}$) (Supplementary Figs. 12 and 13), which restored full virulence to *Focn*Cong:1-1 HS5 (Fig. 5c and Supplementary Fig. 14a). In contrast, Arabidopsis WT was resistant to the other *Focn*Cong:1-1 HS5 transformants that contained only *SIX8* or *FocnCong_v001894* (Fig. 5c and Supplementary Fig. 14a). It should be noted that virulence of knockout mutants that lack the *SIX8-FocnCong_v001894* locus in *Focn*Cong:1-1 was significantly lower than for WT (Fig. 5d and Supplementary Figs. 14–16), suggesting that both *SIX8* and *FocnCong_v001894* are necessary for virulence on Arabidopsis. We therefore designated *FocnCong_v001894* as *Pair with SIX Eight1* (*PSE1*).

**Genetic and functional conservation of the *SIX8* and *PSE1* loci.** Next, we investigated whether the *SIX8-PSE1* pair is conserved in Arabidopsis-infecting *F. oxysporum* isolates. Comparative analysis of highly contiguous and available genome assemblies of *F. oxysporum* isolates (Supplementary Table 1) showed that the *SIX8-PSE1* locus is completely conserved in Fo5176 and in the stock-infecting pathogen *F. oxysporum* f. sp. *matthiolae* (*Fomt*) PHW726, which can infect Arabidopsis[13,30], but not in isolates that cannot infect Arabidopsis (Fig. 6a, b). For example, the banana-infecting pathogen *F. oxysporum* f. sp. *cubense* (*Focb*)

tropical race 4 (TR4), which threatens banana production worldwide, has *SIX8* but not *PSE1*. In the other non-Arabidopsis-infecting isolates, except *Fol*4287, neither *SIX8* nor *PSE1* is present. *Fol*4287 has multiple copies of *SIX8* and its homolog *SIX8b*[12,31] but *PSE1* is not present in the published *Fol*4287 gene annotation[6]. However, we found three loci similar to the *SIX8-PSE1* locus in chromosomes 2, 3, and 14 of *Fol*4287. At these loci, adjacent to *SIX8*, there is a *PSE1*-like gene (*PSL1*) differing in the C-terminal 10 amino acids (Fig. 6b and Supplementary Fig. 17). Furthermore, multiple *SIX8b* loci contain TEs inserted into adjacent *PSE1* sequences. For example, a transposase gene was found in the first intron of the *PSE1* homologs in two loci of chromosome 3 and another locus in chromosome 6 (Fig. 6b and Supplementary Fig. 18). Similarly, a presumptive transposase was found immediately upstream of the potential-but-unannotated *PSE1* homolog in another locus in chromosome 6 (Fig. 6b). Thus, TE insertion seems to have disrupted the *PSE1* adjacent to *SIX8b* in *Fol*4287.

To evaluate if the *SIX8-PSE1* locus in *Fomt*PHW726 and the *SIX8-PSL1* locus in *Fol*4287 are able to function similarly in *Focn*Cong:1-1, we cloned these loci and transformed them into the *Focn*Cong:1-1 HS5 mutant (ΔSC9/chr$^{SC10/SC20}$; Supplementary

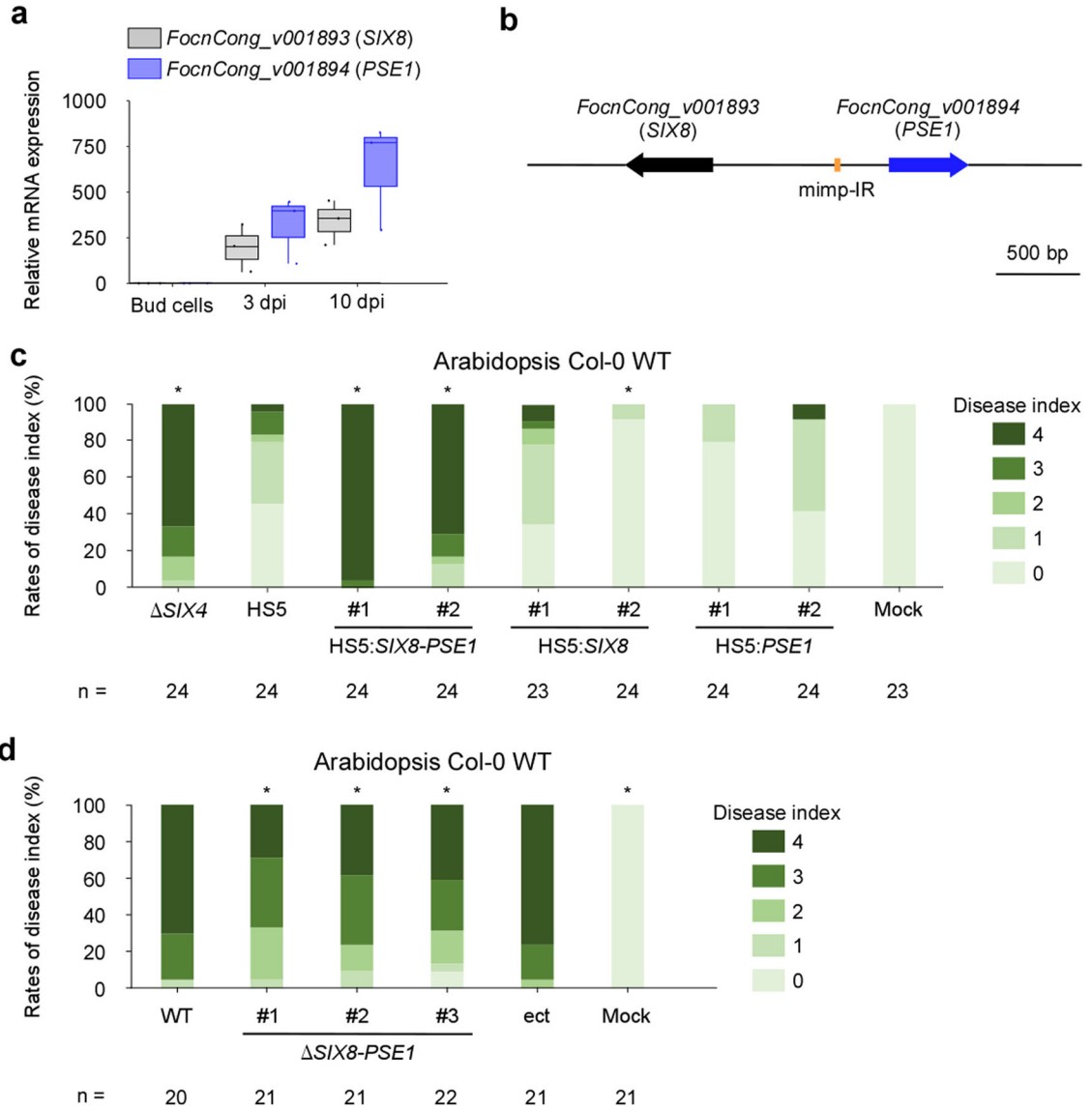

**Fig. 5 The *SIX8-PSE1* locus is involved in virulence of *Focn*Cong:1-1 on Arabidopsis. a** Relative transcript levels of *FocnCong_v001893* (*SIX8*) and *FocnCong_v001894* (*PSE1*) during infection of Arabidopsis Col-0 WT at 3 and 10 dpi. Data from three biologically independent samples are presented as fold changes compared with expression levels in bud cells. Expression levels were determined by qRT-PCR and normalized against *Focn*Cong:1-1 β-tubulin (*TUB2*). Boxplots indicate median value, estimated 25th and 75th percentiles, and whiskers represent 1.5 times the interquartile range. **b** Schematic representation of the *SIX8-PSE1* locus in *Focn*Cong:1-1. mimp-IR miniature impala-like inverted repeat. **c** Virulence of *Focn*Cong:1-1 Δ*SIX4*, HS5, and HS5 transformants introduced with *SIX8* (HS5:*SIX8*), *PSE1* (HS5:*PSE1*) or both (HS5:*SIX8-PSE1*) on Arabidopsis Col-0 WT. Disease index was scored as described in Methods. *n* denotes the number of plants investigated. Asterisks represent significant difference from *Focn*Cong:1-1 HS5. (*$p < 0.001$, Mann–Whitney U-test). Representative images of Arabidopsis at 28 dpi are shown in Supplementary Fig. 14a. **d** Disease index of Arabidopsis Col-0 WT challenged with *Focn*Cong:1-1 WT, Δ*SIX8-PSE1*, an ectopic transformant (ect) or water (mock) at 28 dpi was scored as described in Methods. Results of three independent experiments were combined. *n* denotes the number of plants investigated. Asterisks represent significant difference from WT (*$p < 0.05$, Mann–Whitney U-test). Representative images of Arabidopsis at 28 dpi are shown in Supplementary Fig. 14b.

Figs. 19 and 20). Transformation with the *Fomt SIX8-PSE1* locus, but not the *Fol SIX8-PSL1* locus, restored full virulence to *Focn*Cong:1-1 HS5 in Arabidopsis Col-0 WT (Fig. 6c and Supplementary Fig. 21). These results suggest that the *SIX8-PSE1* locus is functionally distinct from *SIX8-PSL1* and is functionally conserved in Arabidopsis-infecting *F. oxysporum* isolates.

## Discussion

Here we report the identification of a CD chromosome in *F. oxysporum* that is required for virulence on Arabidopsis. This CD chromosome encodes a pair of effectors (SIX8 and PSE1) that are

involved in suppressing Arabidopsis-specific immunity, and are conserved in the other *F. oxysporum* isolates capable of infecting Arabidopsis. The mode of action potentially involves defense against, or suppression of, the phytoalexin camalexin. We also report that another CD chromosome is required for pathogenicity on cabbage. In addition, certain CD chromosomes are involved in conidial formation.

In plant pathogenic fungi, CD chromosomes associated with virulence are usually not involved in vegetative growth[1,2]. In this sense, SC3 and chr$^{SC10/SC20}$ in *Focn*Cong:1-1 are atypical CD chromosomes that affect conidial formation (Fig. 2c). Although the reduced virulence of SC3-deficient *Focn*Cong:1-1 mutants

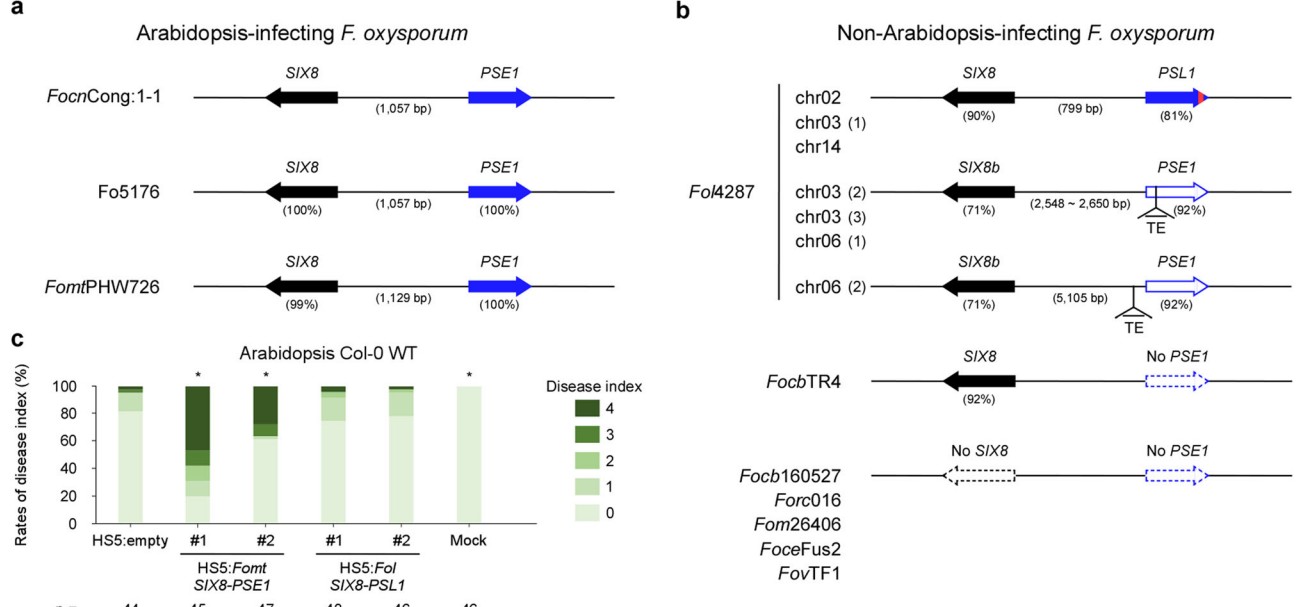

**Fig. 6 Genetic and functional conservation of a pair of effectors, SIX8 and PSE1, among F. oxysporum isolates.** Comparison of the SIX8-PSE1 loci in Arabidopsis-infecting (**a**) and non-Arabidopsis-infecting (**b**) F. oxysporum isolates. A red region in the arrowhead of PSL1 indicates a region of 10 amino acids that differ from PSE1. The amino acid level identity (%) to FocnCong:1-1 SIX8 or PSE1 is shown in parentheses. FocnCong:1-1, F. oxysporum f. sp. conglutinans Cong:1-1; FomtPHW726, F. oxysporum f. sp. matthiolae PHW726; Fol4287, F. oxysporum f. sp. lycopersici 4287; FocbTR4, F. oxysporum f. sp. cubense tropical race 4; Focb160527, F. oxysporum f. sp. cubense 160527; Forc016, F. oxysporum f. sp. radicis-cucumerinum 016; Fom26406, F. oxysporum f. sp. melonis 26406; FoceFus2, F. oxysporum f. sp. cepae Fus2; FovTF1, Fusarium oxysporum f. sp. vasinfectum TF1. **c** Virulence of FocnCong:1-1 HS5 transformants introduced with a hygromycin B resistance gene (hph) cassette (HS5:empty), the Fomt SIX8-PSE1 locus (HS5:Fomt SIX8-PSE1) and the Fol SIX8-PSL1 locus (HS5:Fol SIX8-PSL1) on Arabidopsis Col-0 WT. Disease index was scored as described in Methods. Results of six independent experiments were combined. n denotes the number of plants investigated. Asterisks represent significant difference from FocnCong:1-1 HS5:empty. (*p < 0.01, Mann–Whitney U-test). Representative images of Arabidopsis at 28 dpi are shown in Supplementary Fig. 21.

(HS2, HS3, and HS4) on Arabidopsis and cabbage (Fig. 2c, d) may be due to deficiency in the ability to form conidia, or to regulatory step(s) that has multiple unexplored phenotypic effects, we cannot exclude the possibility that yet-unknown effectors located on SC3 are implicated in virulence. Interestingly, SC3 contains a region partly syntenic to chromosome 11, which is a core chromosome of Fol4287 (Fig. 1a). This syntenic region may contain dose-effective genes involved in conidial formation. In contrast to SC3, chr$^{SC10/SC20}$ negatively regulates conidial formation but positively contributes to virulence on Arabidopsis but not on cabbage (Fig. 2c, d), possibly representing a trade-off between vegetative growth and virulence to a particular host.

FocnCong:1-1 carries multiple CD chromosomes that have distinct virulence functions against specific hosts. For example, the CD chromosome chr$^{SC10/SC20}$-deficient FocnCong:1-1 HS5 is less virulent on Arabidopsis, but is able to develop severe disease on cabbage (Fig. 2d and Supplementary Fig. 5). This result may be explained by the fact that FocnCong:1-1 HS5 maintains the CD chromosome SC8, which harbors a gene, SIX1, required for full virulence on cabbage[14]. Consistently, FocnCong:1-1 HS6, which lacks both SC8 and chr$^{SC10/SC20}$, lost pathogenicity on both cabbage and Arabidopsis (Fig. 2d and Supplementary Fig. 5), and introduction of SC8 into HS6 restored virulence on cabbage (Fig. 3c and Supplementary Fig. 9). Thus, we conclude that chr$^{SC10/SC20}$ and SC8 are responsible for host-specific virulence on Arabidopsis and cabbage, respectively.

The target of the CD chromosome chr$^{SC10/SC20}$ effector is likely to be CYP79B2/CYP79B3-mediated immunity in Arabidopsis, because the loss of chr$^{SC10/SC20}$ attenuated virulence of FocnCong:1-1 HS5 to WT, but not to the cyp79b2/cyp79b3 double mutant (Fig. 4a and Supplementary Fig. 10). CYP79B2/CYP79B3 had not previously been implicated in resistance to F. oxysporum.

For instance, Kidd et al.[32] reported that susceptibility of cyp79b2/cyp79b3 to F. oxysporum Fo5176 was not different from WT. Consistent with this report, our study shows that virulence of FocnCong:1-1 on cyp79b2/cyp79b3 is comparable to WT (Fig. 4a and Supplementary Fig. 10). Thus, only the use of CD chromosome-deficient mutants allowed us to uncover the involvement of CYP79B2/CYP79B3 in resistance to F. oxysporum. Furthermore, histological analysis suggests that CYP79B2/CYP79B3-mediated immunity may be associated with inhibition of root–stem translocation of FocnCong:1-1 (Fig. 4b). CYP79B2/CYP79B3 is responsible for synthesis of Trp-derived secondary metabolites, including sulfur-containing compounds that are characteristic of the Brassicaceae[26]. These sulfur-containing antimicrobial compounds differ among Brassicaceae species; for example, camalexin is produced in Arabidopsis, but not in cabbage[28]. Our results suggest that FocnCong:1-1 can overcome the Arabidopsis-specific immunity conferred by PAD3, a camalexin synthetic gene (Fig. 4d and Supplementary Fig. 10c), when the CD chromosome chr$^{SC10/SC20}$ that encodes the paired effectors SIX8 and PSE1 is present. This pair of effectors is highly conserved in Arabidopsis-infecting F. oxysporum isolates, but not in other isolates (Fig. 6), thus the presence of a particular CD chromosome that harbors these effector genes would contribute to the determination of host specificity.

In this study, FocnCong:1-1 HSs were generated by treatment with the mitosis inhibitor benomyl. In the generation process, a genome rearrangement, but not just a chromosome loss, has occurred at least in HS1, HS2, and HS5 (Fig. 2a). We also investigated phenotypes in an additional HS mutant with the same karyotype as HS5 (HS5L: HS5-like mutant; Supplementary Figs. 22 and 23). Like HS5, HS5L showed virulence on cyp79b2/cyp79b3 and pad3 plants, but not on Col-0 WT plants. We cannot

rule out the possibility that these genome rearrangements affect phenotypes. In addition to the results of HS5L, however, the return of HS5 virulence on Arabidopsis in two independent HS5 transformants containing *Focn*Cong1-1 *SIX8-PSE1* (Fig. 5c) supports the conclusion that the *SIX8–PSE1* pair is required for virulence on Arabidopsis.

We identified *SIX8* and *PSE1* as a gene pair adjacent but encoded on opposite DNA strands (head-to-head orientation) (Fig. 5b and Supplementary Fig. 11). Head-to-head orientation of effector genes has been documented for other *SIX* genes in *F. oxysporum*. For instance, in *Fol*, a pair of effector genes *SIX3* (also known as *AVR2*) and *SIX5* are also adjacently located in a head-to-head transcriptional orientation[12,33,34]. Both *SIX3* and *SIX5* are required for not only full virulence in a susceptible host, but also disease resistance in tomato lines containing the resistance gene *I-2*[33–35], and the gene products are thus likely to function as a pair. The close head-to-head orientation may ensure coordinated transcription to produce both proteins at similar levels. Such system would be suitable for maintaining tight stoichiometry of two proteins in a complex. Indeed, SIX5 interacts with SIX3 at plasmodesmata in plant cells, facilitating cell-to-cell movement of SIX3[33,34]. Unlike the SIX3–SIX5 pair, however, we failed to detect direct interaction between SIX8 and PSE1 in a yeast two-hybrid assay (Supplementary Fig. 24). We cannot exclude the possibility that SIX8 indirectly interacts with PSE1, e.g., via host target(s), or the yeast system may not be suitable for detecting interactions of these proteins. Alternatively, SIX8 and PSE1 may act independently. As bioinformatic analysis of SIX8 and PSE1 protein sequences gives no known domain annotations, identification of host targets of SIX8 and PSE1 will be required to clarify functions of the paired effectors. It is also notable that disruption or loss occurs in only *PSE1*, but not in *SIX8*, in certain non-Arabidopsis infecting *F. oxysporum* isolates. Perhaps *PSE1*, but not *SIX8*, is recognizable in plants that carry corresponding resistance proteins, leading to its disruption or loss to avoid detection.

In this work we demonstrate that the host range of *F. oxysporum* depends on CD chromosomes. In this respect, it is interesting that certain isolates, such as *Fol*4287 and *Forc*016, have only a single virulence-associated CD chromosome, whereas *Focn*Cong:1-1 has multiple CD chromosomes, each of which encodes host-specific effectors. Because the *Focn*Cong:1-1 genome is very large (68.8 Mb) compared to most known *F. oxysporum* genomes, such as *Fol*4287 (59.9 Mb)[6] and *Forc*016 (52.9 Mb)[7], *Focn*Cong:1-1 is likely to have expanded its host range by acquiring and maintaining additional CD chromosomes. Indeed, Masunaka et al.[36] have shown that a field isolate of *A. alternata* carrying two putative CD chromosomes has a wide host range. In that case, host-specific toxin genes on different chromosomes determine host range[36]. In the case of *F. oxysporum*, host specificity can be determined, at least in part, by effectors, as seen in this study. Further functional analyses of the *SIX8-PSE1* paired effectors and their derivatives will be needed to dissect out the molecular mechanisms underlying effector-based host specificity in *F. oxysporum*.

## Methods

**Fungal strains and plants**. Fungal strains used in this study are listed in Supplementary Table 2. For pre-incubation, all strains were incubated on potato dextrose agar (PDA; Nissui Pharmaceutical Co.) at 28 °C in the dark. For bud cell production, all strains were grown in $NO_3$ medium (0.17% yeast nitrogen base without amino acids, 3% sucrose and 1% $KNO_3$) at 120 strokes per minute (spm) for 4 days at 28 °C in the dark. For gene expression profiling (Supplementary Data 1 and 2), mycelia were harvested after 10 days of incubation on PDA at 28 °C. Bud cells were collected from $NO_3$ medium by filtration with a nylon mesh and centrifugation. Hyphae trapped with the nylon mesh were collected. Mycelia from PDA, bud cells, and hyphae were stored at –80 °C until RNA isolation.

Arabidopsis (Col-0 wild type, *pen2*, *pad3*, *cyp82c2*, and *cyp79b2/cyp79b3* mutants[26,27]) and cabbage (cv. Shikidori and cv. Shosyu; Takii Seed) were cultured in pots containing autoclaved Super Mix A (Sakata Seed) and vermiculite (VS kakou). Arabidopsis was grown at 22 °C for 10 h under light and 14 h dark in a growth chamber. Cabbage was grown in a greenhouse.

**Bioassays**. For evaluation of disease severity, 14-day-old Arabidopsis and cabbage cv. Shikidori were injured with a forceps or a plastic peg and then irrigated with 1 ml of *Focn*Cong:1-1 bud cell suspension ($1 \times 10^7$ cells/ml). Inoculated Arabidopsis plants were grown at 28 °C for 10 h under light and 14 h dark in a growth chamber. An Arabidopsis disease index was scored at 28 or 29 days post-inoculation (dpi) as: 0, no symptoms; 1, dwarf; 2 yellowing, vein clearing or wilting of one to a few leaves; 3, wilting of a whole plant; 4, dead. A cabbage disease index was also scored at 28 or 29 dpi as: 0, no symptoms; 1, yellowing lower leaves; 2, yellowing lower and upper leaves; 3, whole plant wilting; 4, dead.

For gene expression profiling (Supplementary Data 1 and 2), 20- or 21-day-old Arabidopsis and 17-day-old cabbage cv. Shosyu roots were irrigated with 1 ml of bud cell suspension ($1 \times 10^7$ cells/ml). At 3 dpi and 10 dpi, infected roots were washed with water to remove soil. The roots were stored at –80 °C until RNA isolation.

For observation of colonization of Arabidopsis by *Focn*Cong:1-1, roots of 14-day-old Arabidopsis were cut to approximately 1 cm lengths from the border between roots and stems and soaked in bud cell suspension ($1 \times 10^7$ cells/ml) for 1 min, then transferred to square plates containing soil. At 12 dpi, roots approximately 5 mm below soil surface were observed by an Olympus BX51 epifluorescence microscopy (Olympus) with excitation of 488 nm for GFP. Images were obtained with an Olympus DP74 digital camera (Olympus) and edited with cellSens (Olympus).

**Fungal growth assays**. *Focn*Cong:1-1 strains were grown on PDA for 8 days at 28 °C in the dark from a freezer stock. For measurement of colony diameter, mycelium agar disks were collected from the growing edge of a colony using sterile plastic straws and placed in the center of fresh PDA plates. After 8 days, colony diameter was measured. For quantification of conidial formation, 17-day-old colonies were soaked in 10 ml of water and scraped with a colony spreader. Conidial suspensions were filtrated through a nylon mesh to remove mycelia and conidia were quantified at $OD_{600}$ with a WPA CO 8000 Cell Density Meter (WPA) or by counting using haemocytometer.

**Plasmid construction**. Primers used for plasmid construction are listed in Supplementary Table 3. To generate *SIX8-PSE1* locus complementation vectors, the *Focn*Cong:1-1 *SIX8-PSE1*, *Fomt SIX8-PSE1* and *Fol SIX8-PSL1* loci were amplified from genomic DNAs of *Focn*Cong:1-1, *Fomt*MAFF240332 and *Fol*4287, respectively, and cloned into pCR™8/GW/TOPO® vector using a pCR™8/GW/TOPO® TA Cloning® Kit (Invitrogen) as described by the manufacturer. To introduce these loci into *Focn*Cong:1-1 HS5, the complementation vector containing each locus was co-transformed with pCSN43 containing an *hph* cassette[37]. For transformation vectors of *SIX8* or *PSE1*, an *hph* cassette was amplified from pCSN43[37] and assembled with *SIX8* or *PSE1*, which was amplified from the *Focn*Cong:1-1 *SIX8-PSE1* locus complementation vector as a template, using NEBuilder HiFi DNA Assembly Master Mix (New England Biolabs) as recommended by the manufacturer. The assembled fragments were cloned into pCR™8/GW/TOPO®.

To generate the *Focn*Cong:1-1 *SIX8-PSE1* locus disruption vector, the flanking regions of *SIX8* and *PSE1* were amplified from the *Focn*Cong:1-1 *SIX8-PSE1* locus complementation vector as a template and assembled with an *hph* cassette using NEBuilder HiFi DNA Assembly Master Mix (New England Biolabs). The assembled fragment was cloned into pCR™8/GW/TOPO®.

Constructs for yeast two-hybrid assays were generated from cDNAs of *SIX8* and *PSE1* without signal peptide sequences or a stop codon by amplification from cDNA generated from mRNA isolated from *Focn*Cong:1-1-infected Arabidopsis. Amplicons were inserted into pENTR™/D-TOPO® (Invitrogen), and then into yeast expression vectors pDEST-DB and pDEST-AD[38] using Gateway™ LR Clonase™ II Enzyme Mix (Invitrogen) as described by the manufacturer.

**Protoplast formation and transformation**. For creation of GFP-labeled *Focn*Cong:1-1 strains (Δ*SIX4*-GFP, HS2-GFP, and HS5-GFP), HS5 transformants (HS5: *SIX8-PSE1*, HS5:*SIX8*, HS5:*PSE1*, HS5:*Fomt SIX8-PSE1* and HS5:*Fol SIX8-PSL1*) and *SIX8-PSE1* knockout mutants (Δ*SIX8-PSE1*), which are listed in Supplementary Table 2, bud cells ($4 \times 10^8$) were incubated in 80 ml of potato dextrose broth (Difco) at 80 spm for 16 h at 28 °C. Mycelia germinated from the bud cells were collected by centrifugation (1800 *g*, 10 min) and washed with 1.2 M $MgSO_4$. Mycelial cell walls were digested with 25 ml 2% (w/v) Driselase (Sigma) and 2% (w/v) Lysing Enzymes (Sigma) in 1.2 M $MgSO_4$, and maintained at 80 spm for 3 h at 28 °C. Protoplasts were collected by filtration with a nylon mesh and centrifugation (1500 *g*, 10 min), and rinsed twice with 0.7 M NaCl. The protoplasts were resuspended in STC (1.2 M sorbitol, 50 mM $CaCl_2$, 10 mM Tris–HCl pH 7.5) and adjusted to $1 \times 10^8$ cells/ml. For polyethylene glycol transformation, 30 μg plasmid DNA was added to 150 μl of the protoplast suspension as previously described[39]. Transformants were selected and maintained on PDA containing

hygromycin B (100 µg/ml) or G418 (200 µg/ml) and verified by PCR using primers listed in Supplementary Table 3. Plasmid DNAs used for transformation are shown in Supplementary Table 4.

**Genome sequencing and assembly**. For PacBio sequencing, genomic DNA of *Focn*Cong:1-1 was isolated using CTAB and 100/G genomic tips (QIAGEN) as described in the 1000 Fungal genomes project (http://1000.fungalgenomes.org). The genome was sequenced on five PacBio RSII cells and assembled by the Hierarchical Genome Assembly Process (HGAP) v4 within SMRT Link (v5.1.0). Default values were kept and the expected genome size was set to 70 Mb.

For optical mapping, genomic DNA was isolated using a Blood and Cell Culture DNA Isolation Kit (Bionano Genomics) as described by the manufacturer. Genomic DNA was labeled with an NLRS Labeling Kit (Bionano Genomics) with *Bsp*QI and *Bbv*CI as described by the manufacturer. The labeled DNA was scanned using a Bionano Irys platform. Bionano maps from two enzymes (*Bsp*QI and *Bbv*CI) (Bionano Solve v3.2) were merged with PacBio sequence assemblies to produce long hybrid scaffolds. Completeness of gene space within the assembly was assessed through the presence of conserved single-copy genes using BUSCO version 3.0.2[40,41]. Analysis with the Sordariomyceta data set (3725 genes) indicated the presence of 3690 genes (99.1%) in the assembly (Table 1). Whole-genome alignments were performed with nucmer (with –maxmatch) in MUMmer 3.23[42].

For genome sequencing of *Focn*Cong:1-1 ΔSIX4 and HSs, genomic DNA was isolated using DNeasy Plant Mini Kits (QIAGEN). Illumina NovaSeq 6000 or HiSeq 2500 paired-end sequencing was used for *Focn*Cong:1-1 ΔSIX4 and HSs, except for HS3, using a library with a mean insert size of 550 bp. Illumina NextSeq 500 single-end sequencing was used for *Focn*Cong:1-1 HS3, from library preparation with a mean insert size of 350 bp. The Illumina sequence library was quality-filtered using the FASTX Toolkit 0.0.13.2 (Hannonlab) with parameters -q20 and -p50. Reads containing "N" were discarded. Quality-filtered libraries were aligned with the *Focn*Cong:1-1 genome using CLC Genomic Workbench 20 using default settings.

**RNA extraction, cDNA synthesis, and qRT-PCR**. Total RNAs were extracted using RNeasy Plant Mini Kit (QIAGEN). Total RNA (200–1000 ng) was used to generate cDNA in a 20 µl volume reaction according to the Invitrogen Superscript III Reverse Transcriptase protocol. cDNA was diluted 1:5, and 1 µl was used for a 10 µl qPCR reaction with 5 µl THUNDERBIRD SYBR Green mix (Toyobo) on an Mx3000P qPCR System (Agilent) using the following program: (1) 95 °C, 1 min, (2) [95 °C, 15 s, then 53 °C, 30 s, then 72 °C, 1 min] × 40, (3) 95 °C, 1 min for SIX8 and PSE1, or (1) 95 °C, 1 min; (2) [95 °C, 15 s, then 60 °C, 30 s, then 72 °C, 1 min] × 40, (3) 95 °C, 1 min for *Focn*Cong:1-1 β-tubulin (TUB2), followed by a temperature gradient from 55 to 95 °C. Standard curves were generated using serial dilutions of cDNAs from Arabidopsis infected with *Focn*Cong:1-1 at 10 dpi for SIX8 and PSE1 and cDNAs from bud cells for *Focn*Cong:1-1 TUB2. *Focn*Cong:1-1 TUB2 was used as a reference gene. Primers used for qPCR are listed in Supplementary Table 3.

**RNA sequencing**. Using the extracted RNA, strand-specific shotgun type of RNA library was prepared using the Breath Adapter Directional sequencing protocol[43]. Briefly, mRNA was extracted and fragmented using magnesium ions at elevated temperature. The polyA tails of mRNA was primed by an adapter-containing oligonucleotide for cDNA synthesis. 5′ adapter addition was performed by breath capture technology to generate strand-specific libraries. The final PCR enrichment was performed using oligonucleotides containing the full adapter sequence with different indexes. After cleanup and size selection, concentration of library was measured by microplate photometer Infinite® 200 PRO (TECAN) to pool libraries for Illumina sequencing systems. The libraries were sequenced on an Illumina NextSeq 500 platform. The Illumina sequence library was quality-filtered and aligned as above. Transcription levels for each transcript were calculated as TPM (transcripts per million).

**Gene prediction and annotation**. RNA sequencing data from *Focn*Cong:1-1 was aligned with the *Focn*Cong:1-1 genome using HISAT2 v.2.1.0[44] and used to guide gene model prediction using the BRAKER1 v1.9 pipeline[45]. BRAKER1 was run with the repeat-softmasked genome created by RepeatMasker v.4.0.7 (with -engine ncbi -species "ascomycota" –xsmall; http://www.repeatmasker.org/), using the fungal and softmasking options. Gene-coding sequences were annotated through BLASTp (E-value cutoff = 1E-6) searches against the July 2018 release of the SWISS-PROT database[46]. Putative secreted proteins were identified through prediction of signal peptides using SignalP v.5.0[47] and removal of sequences with TMHMM v.2.0[48]-predicted transmembrane domains. For effector prediction, putative secreted proteins were screened for proteins with an effector-like structure using EffectorP 1.0 and/or 2.0[17,18]. In addition, BLASTp analyses (E-value cutoff = 1E-6) were performed for the fourteen SIX genes (SIX1-SIX14) and the four genes (FOA1-FOA4) known to be effectors in Arabidopsis-infecting *F. oxysporum*[12,16].

**Analysis of repeat elements**. Repeat element prediction was performed using the genome sequences of eight *F. oxysporum* strains in the NCBI database that had contig N50 values greater than 1 Mb (last accessed on November 24, 2019) as described in Gan et al.[49]. Code used for this analysis is available at: https://github.

com/pamgan/colletotrichum_genome. The details of genome sequences used for this analysis are shown in Supplementary Table 5. Briefly, repeat sequences were predicted using RECON and RepeatScout via RepeatModeler open-1.0.11 (http://www.repeatmasker.org), TransposonPSI (http://transposonpsi.sourceforge.net/), LTR_retriever[50], and LTRPred[51] (https://github.com/HajkD/LTRpred). Sequences that were longer than 400 bp from TransposonPSI, LTR_retriever, and LTRPred were combined and used as queries for BLASTx against RepBase[52] peptide sequences bundled in RepeatMasker open-4.0.9-p2 (http://www.repeatmasker.org). Lastly, these sequences were used as queries for BLASTn against each fungal genome. Only sequences with more than five hits (BLASTn E-value cutoff = 1E-15) and/or with a hit to a RepBase peptide (BLASTx E-value cutoff = 1E-5) were retained for further analysis. Sequences from all sources were combined using VSEARCH v2.14.0[53], using 80% identity as the cutoff threshold. Consensus sequences were classified using RepeatClassifier (from RepeatModeler open-1.0.11). Known *Fusarium* repeat sequences registered in Dfam_Consensus-20181026 and RepBase-20181026 were extracted, except for those that were annotated as artefacts, simple repeats, or low complexity sequences. The custom repeat library was created by combining the consensus sequences and known *Fusarium* repeat sequences, and used as input for RepeatMasker open-4.0.9-p2. The "one code to find them all"[54] was used to reconstruct repeat elements.

**Chromosome loss and transfer**. A chromosome loss experiment was performed according to VanEtten et al.[22]. *Focn*Cong:1-1 ΔSIX4 was incubated in M100 medium (1% glucose, 0.3% KNO₃, 6.25% salt solution) with benomyl (1.56, 3.13, or 6.25 µg/ml) at 120 spm for 4 days at 28 °C. The salt solution consists of 0.4% KH₂PO₄, 0.4% Na₂SO₄, 0.8% KCl, 0.2% MgSO₄·2H₂O, 0.1% CaCl₂, and 0.8% trace elements (0.006% H₃BO₃, 0.014% MnCl₂·4H₂O, 0.0844% ZnSO₄·7H₂O, 0.004% NaMoO₄·2H₂O, 0.006% FeCl₃, 0.04% CuSO₄·5H₂O). Hyphae were removed with a nylon mesh, and bud cells were collected by centrifugation at 1630 g for 10 min. Supernatant was discarded and the remnant with bud cells was spread on M100 plates containing 2% agar and 0.04% Triton X-100 (Wako), and the inoculated plate was overlaid with an autoclaved filter paper. Plates were incubated at 28 °C for 1 to 3 days, then the filter paper was transferred onto M100 medium containing hygromycin B (100 µg/ml) and incubated at 25 °C overnight. Hygromycin B-sensitive isolates were selected by comparing the plates, and then chromosome loss patterns were verified by PCR (Supplementary Fig. 2) using primers listed in Supplementary Table 3.

Chromosome transfer experiments were performed according to van der Does and Rep[55]. A zeocin-resistant *Focn*Cong:1-1 HS6 (HS6-BLE) strain was generated by *Agrobacterium*-mediated transformation as previously reported[56] with *Agrobacterium tumefaciens* EHA105 harboring pRW1p[57]. *Focn*Cong:1-1 ΔSIX4 and HS6-BLE were co-incubated on PDA at 25 °C. Conidia were harvested from 7-day-old colonies, and conidial suspensions were spread on PDA containing hygromycin B (100 µg/ml) and phleomycin (100 µg/ml). Double drug-resistant colonies were selected, and then chromosome patterns were verified by PCR (Supplementary Fig. 6) using primers listed in Supplementary Table 3.

**Contour-clamped homogeneous electric field (CHEF) gel electrophoresis**. CHEF gel plugs were made by resuspending protoplasts in STE (1 M sorbitol, 25 mM Tris-HCl pH 7.5, 50 mM EDTA). Protoplast concentration was adjusted to 4 × 10⁸ cells/ml and added to the same amount of 1.2% low melting agarose gel (Bio-Rad) solution. Protoplast suspensions (2 × 10⁸ cells/ml) containing 0.6% low melting agarose gel were added to 50-well dispensable mold plates (Bio-Rad). Plugs were immersed in 10 ml of NDS (1% N-lauroyl sarcosine sodium salt solution, 0.01 M Tris-HCl, 0.5 M EDTA) and incubated at 65 spm for 14 to 20 h at 37 °C. NDS was replaced with 0.05 M EDTA three times every 30 min. Plugs in 0.05 M EDTA were stored at 4 °C until use.

CHEF gel electrophoresis was done according to Inami et al.[58]. Briefly, chromosomes were separated on 1% SeaKem® Gold Agarose (Lonza) in 0.5×TBE buffer at 4 to 7 °C for 260 h using a CHEF Mapper System (Bio-Rad). Switching time was 1200 to 4800 s at 1.5 V/cm with an included angle of 120°. The running buffer was exchanged every two or three days. Chromosomes of *Hansenula wingei* (Bio-Rad) were used as a DNA size marker. Gels were stained with 3×GelGreen (Biotium) to visualize chromosomes.

**Yeast two-hybrid assays**. For yeast two-hybrid assays, bait (pDEST-DB; DB) and prey vectors (pDEST-AD; AD) containing cDNA of SIX8, PSE1 or empty vector controls were transformed into *S. cerevisiae* Y8930 and Y8800, respectively, with a slight modification of the method described by Lopez and Mukhtar et al.[59]. Transformants carrying DB and AD were selected with synthetic defined (SD) media (0.67% yeast nitrogen base, 0.5% glucose, 0.01% adenine hemisulfate salt) supplemented with -Leu DO supplement (Clontech) (SD-Leu) and -Trp DO supplement (Clontech) (SD-Trp), respectively. Yeast transformants were mated by yeast extract peptone dextrose growth broth (1% yeast extract, 2% peptone, 2% glucose, 0.01% adenine hemisulfate salt) at 150 spm for 24 h at 28 °C. Diploid cells were selected with SD supplemented with -Leu/-Trp DO supplement (Clontech) (SD-Leu-Trp), and spotted on SD supplemented with -Leu/-Trp/-His DO supplement (Clontech) (SD-Leu-Trp-His) and SD-Leu-Trp with 1 mM 3-amino-1,2,4-triazole. Yeast colonies were observed after 72 h incubation.

**Statistics and reproducibility**. All statistical analyses were performed in EZR[60]. Welch's *t*-test was used to analyze the statistical significance for continuous variables (e.g., $OD_{600}$ value of conidial suspensions), whereas Mann–Whitney *U*-test was used for evaluation of disease severity. The reproducibility was determined by using independent biological replicates as indicated in the figure legends. Individual values for data plots are included in Supplementary Data 3.

**Reporting summary**. Further information on research design is available in the Nature Research Reporting Summary linked to this article.

## Data availability

The Whole Genome Shotgun project of *Focn*Cong:1-1 has been deposited at DDBJ/ENA/GenBank under the accession RSAI00000000 (BioProject number PRJNA506492 and BioSample number SAMN10461798). The version described in this paper is version RSAI01000000. RNA sequencing data from culture medium and plant infections have been deposited in NCBI's Gene Expression Omnibus (GEO) and are accessible through GEO Series accession number GSE157823. The source data underlying Fig. 2c, d, 3b, c, 4a, b, d, 5a, c, d and 6c are provided as Supplementary Data 3. Other data are available by reasonable request.

## Code availability

Code for repeat element prediction are available at: https://github.com/pamgan/colletotrichum_genome.

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

## Acknowledgements

We thank Prof. Yoshitaka Takano (*pen2* and *pad3*), Dr. Elizabeth S. Sattely (*cyp82c2*), and Dr. Kei Hiruma (*cyp79b2/cyp79b3*) for providing seeds. We also thank Dr. M. Shahid Mukhtar for providing pDEST-DB and pDEST-AD vectors and Dr. Kazuki Sato for technical assistance in yeast two-hybrid assays. We would also like to thank Prof. Hiroyuki Kasahara for fruitful discussions. This work was supported by JSPS KAKENHI 19H00939 (S.A. and T.A.), 20H02995 (S.A.), 17K07679 (S.A.), 19K21154 (Y.A.), and 17H06172 (K.S.); JST PRESTO Grant Number JPMJPR16O1 (S.A.); the Institute for Fermentation, Osaka (Y.A.); research fellowship from the Japan Society for the Promotion of Science (Y.A.).

## Author contributions

Y.A., S.A., P.G., A.T., I.Y., and A.S. conducted experiments. Y.A., S.A., K.K., P.M.H., M.R., K.S., and T.A. conceived and supervised the study. Y.A., S.A., K.S., and T.A. wrote the manuscript. All authors reviewed and approved the manuscript.

## Competing interests

The authors declare no competing interests.
