## [Peer Review File · Communications Biology]

Referee expertise:

Referee #1: genomics, fungal pathogens

Referee #2: phytopathogens

Reviewers' comments:

Reviewer #1 (Remarks to the Author):

Review Yu et al. 2020

Fusarium oxysporum isolates have been shown to harbor lineage-specific conditionally dispensable chromosomes (CD) which can be transferred between strains and encode effector genes important for virulence. Here, the authors report the genome sequence of a *Fusarium oxysporum* isolate (*F. oxysporum* f. sp. *conglutinans* (Focn) that is virulent on the model plant *Arabidopsis thaliana* and on cabbage. By isolating several derivatives of this strain lacking different sets of such dispensable chromosomes/scaffolds the authors show that Focn harbors at least 5 dispensable chromosomes. More importantly, they convincingly show that different CD chromosomes are important for conferring virulence on the different hosts. While two of these (SC5 and/or SC8) are important to confer virulence on cabbage, the CD chromosome chr.16 (SC10/20) encodes functions specifically required for virulence on *Arabidopsis*. The authors also provide evidence that the two highly expressed and neighboring effector genes SIX8 and PSE1 from chr.16 (SC10/20) play central roles in this suppression and are conserved amongst strains virulent on *Arabidopsis*.

A very interesting paper on conditionally dispensable mini chromosomes and their important functions for virulence in fungi that should be of considerable interest also to people outside the field. While the observation that dispensable chromosomes can be specific for virulence has been made years ago, the genomic era allows for a completely novel and intriguing look at this phenomenon. The data reported are well documented, highly relevant and most of the conclusions well justified. However, the manuscript is not without flaws (see below) that should be addressed. I very much enjoyed reading the manuscript although parts of it are rather difficult and some of the arguments unnecessarily complicated and twisted.

Major points:

1. One of the claims repeatedly made in the ms, including the abstract, is that CD chromosome SC3 and in part chr.16 (SC10/20) affect vegetative growth (line154, line 179 and elsewhere). This is an over interpretation and misleading because conidiation is analyzed, not growth rate. In addition, conidiation is not carefully analyzed but inferred from simple OD measurements. And in the end, these findings are not essential for the main arguments. This must be either more carefully studied or left out of their arguments, particularly in the abstract. As it stands, it makes the interpretations less convincing.
2. The ms would be much easier to read if the authors had adopted the chromosome assignments of the recently published and highly similar genome of Fo5176, instead of referring to their own scaffolds of an incomplete assembly. This would also make Figure 2b and Figure 3a so much easier to understand.
3. None of the pictures of affected plants is sufficiently detailed to be evaluated by the reader. The authors should provide larger pictures in the supplement.

Additional points:

Line 67: Maybe a rephrasing here could improve understanding the details.

For example by stating how SIX6 differs between the species.

Line 93: maybe add a sentence to clarify what the situation is with the other SIX genes not mentioned here? Only the insiders would know how many there are.

Line 98 and Supplemental data 1: The authors report on 265 secreted effectors, but in their Supplementary data 1 some appear to be identical e.g. FocnCong_v001238 and Focn_v001293. This should be clarified.

Supplementary data 1: Please check spelling in footnotes; Why do you use the term "Target gene"?

Line 110: Please add quantitative data concerning TE content.

Supplement data 2: Please check the formatting of commas and dots. In other words, why are the data not rounded? The authors should provide a complete list of expression data not just the effectors. Only then can expression of the selected genes here be compared to the expression of the core genome.

Line 119: Here it seems at least to me that the fact that the genome reported here and that of isolate Fo5176 (reference 21) are >95 % identical might raise potential questions concerning the novelty of the genomic data. The authors should comment or at least they should discuss the differences between these two genomes more intensively here or elsewhere. For example how similar are the coding regions? Which genes are missing?

Line 127: This is a critical experiment and a paragraph hard to read!

Benomyl treatment is obviously mutagenic, therefore the frequencies of chromosome loss should be pointed out here to judge how frequently these genotypes arise. The authors should also give some information as to whether or not additional rearrangements or mutations other than chromosome loss can be observed (The authors have the data to do this). In addition, the pulsed field gel of HS2 in figure 2 suggests that a genomic rearrangement has occurred here, not just a chromosome loss. Please comment.

The authors analysed only one strain for each karyotype. As a consequence, their conclusions regarding the function of particular CD chromosomes can be tricky. I understand that all science is the art of the doable, but I would be more careful here. Many additional mutations may occur that affect the phenotype that can easily be overlooked.

Maybe the authors can include more than just one mutant for the critical karyotype HS5.

Maybe, cell sorting might do the job to generate these as here no mutagenic material is used. The authors should also provide a list of genes missing in the respective mutants HS1 to HS6. As they have the data, this should be easy.

Line 142 and Figure 2: line 975:

I am afraid that with such small pictures I cannot judge the course of infection. Full size pictures should be provided in the supplement.

Line 144: given that the assay used to measure conidiation is only based on OD measurements this may be an overstatement. Showing the plates (supplement) is fine but biomass could be measured more quantitatively. Conidiation would only be a concern if conidiation is required for virulence. The effect on conidiation could be more intensively studied to support the strong statement in their abstract that the authors have identified an unusual CD chromosome affecting vegetative growth. Microscopy and counting the conidia may be helpful. The authors imply that loss of conidiation affects

virulence, but the amount of inoculated bud cells is the same in all cases, so this, in my opinion, should not matter.

Line 151-254: The conclusion here is hard to follow:

There is no evidence that SC10/20 is associated with vegetative growth other than increased numbers of conidia.

Line 157: Where was HS6 is tagged with phleomycin? If they know they should include this.

Line 188:

In this paragraph it is not clear why the authors did not analyze strains HCT2, HCT3 and HCT4 because those would have been nice controls for the observed effects. This should be included. I also could not find any information about SIX4-GFP or about HS2-GFP. Maybe include were to find it.

Line 221: the pad3 mutant was more susceptible to HS5. The authors may in the future consider analysing cyp71A27 as it is active in roots.

Line 229 the pair of effectors:

The transcription starts which can be deduced from their RNA Seq analysis should be included here, in order to get an idea about the distance of the promoters or the presence of a bidirectional promoter.

Line 242: complementation of HS5. While the data support the notion that both genes SIX8 and PSE1 are required for the biological effect, it cannot be excluded that in the strains harboring only SIX8 or PSE1 the lack of complementation is due to a lack of expression. So, it is still possible that only one of the two genes is required. Since this a strong statement in their title it might be a good idea to check expression here.

Line 245: The effects of deleting the locus is an important experiment. Did it have to go to the supplement because the effect was so small?

Line 250: here the authors should provide more details concerning the conservation of SIX8 and PSE1 among Arabidopsis infecting oxysporum species, including either a sequence alignment or a quantitation (% identity). How for example does completely conserved differ from conserved?

Line 250: Very nice to show that SIX8-PSE1 locus is active from Arabidopsis infecting strain but not from a strain lacking virulence on Arabidopsis!

Line 323: The sequences should be compared here or elsewhere with regard to protein domains. Even if the protein sequences of SIX8 and PSE1 may not be particularly informative, the authors should say something about it, as the reader will be asking this.

Line 342: Please rephrase.

Summary: Nice work.

Reviewer #2 (Remarks to the Author):

This is an informative and well-reasoned manuscript describing the genomic features of a strain of *Fusarium oxysporum* that causes disease on cabbage and Arabidopsis. The authors sequence strain Cong 1-1 and compare it to genomes of other strains with the same or different host specializations. They identify features consistent with conditionally dispensable (CD) chromosomes and these as confirmed to be CD by mutagenesis studies. By testing various mutants they narrow down genomic scaffolds that, when missing, alter pathogenic phenotype. One such scaffold (SC10/SC20) is found to be required for virulence on Arabidopsis but not cabbage. Arabidopsis resistance is dependent upon the plant gene PAN3 involved in the synthesis of the phytoalexin camalexin thus the SC10/SC20 conferred virulence appears to suppress camalexin resistance in some manner. Two adjacent genes (SIX8 and PSE1) for predicted effectors found on SC10/SC20 when added together restore virulence when added back to a strain lacking the entire SC10/SC20. Neither gene alone restores virulence. Overall, this is an impressive piece of work describing the genomic features of this model phytopathogen and identifying specific genes and other regions of the genome of significance to pathogenicity. Other than described below, all major claims of the paper are justified. This will add to our knowledge of this important pathosystem.

My only comment and suggestion for improvement involves the section describing how fungal mutations affect infection phenotype (Figure 4b). First, the photos illustrating the phenotypes are very hard to see because they are so small. Can there be a supplementary figure showing the phenotypes in more detail and explaining exactly how the comparisons were conducted? This is not described in much detail in the Methods section. Because scoring of the phenotypes seem somewhat subject, were these phenotypic assessments conducted blind without knowledge of the genotype of the sample being examined? In the Figure, all comparisons were made to the deltaSIX4-GFP / Col-0 control which includes differences in both the plant and pathogen genotypes. But perhaps the more important comparison is the difference for the mutant HS5-GFP on Col-0 versus cyp79b2/cyp79b3. Can this be included?

Thank you very much for your helpful comments and critique of our manuscript (COMMSBIO-20-3089-T). We fully revised the manuscript in line with your comments, and described the points in the text.

First of all, we would like to explain about the changes that were not pointed out by Editor and Reviewers.

- New Figure 5a:

We had used the primer set (FoTEF-Q2-F/R) reported by van der Does et al. (2008) to amplify *Focn EF1 α* as an internal control for qPCR. However, now we found that the primer set amplifies a band even from Arabidopsis-only sample. Therefore, we designed new primers (552 & 553) to amplify *Focn* β -tubulin (*TUB2*) gene as a control and reanalysed expression levels of *SIX8* and *PSE1* by qPCR, renewing the results of new Figure 5a. The primer sequences were shown in new Supplementary Table 3.

- New Supplementary Figure 2b (previous Supplementary Figure 1b):

Since the previous electrophoretic photograph contained cropped data, the same experiment was performed and the new electrophoretic photograph replaced the old one.

Response to the comments of **Reviewer #1**

Fusarium oxysporum isolates have been shown to harbor lineage-specific conditionally dispensable chromosomes (CD) which can be transferred between strains and encode effector genes important for virulence. Here, the authors report the genome sequence of a *Fusarium oxysporum* isolate (*F. oxysporum* f. sp. *conglutinans* (*Focn*) that is virulent on the model plant *Arabidopsis thaliana* and on cabbage. By isolating several derivatives of this strain lacking different sets of such dispensable chromosomes/scaffolds the authors show that *Focn* harbors at least 5 dispensable chromosomes. More importantly, they convincingly show that different CD chromosomes are important for conferring virulence on the different hosts. While two of these (SC5 and/or SC8) are important to confer virulence on cabbage, the CD chromosome chr.16 (SC10/20) encodes functions specifically required for virulence on *Arabidopsis*. The authors also provide evidence that the two highly expressed and neighboring effector genes *SIX8* and *PSE1* from chr.16 (SC10/20) play central roles in this suppression and are conserved amongst strains virulent on *Arabidopsis*.

A very interesting paper on conditionally dispensable mini chromosomes and their important functions for virulence in fungi that should be of considerable interest also to people outside the field. While the observation that dispensable chromosomes can be specific for virulence has been made years ago, the genomic area allows for a completely novel and intriguing look

at this phenomenon. The data reported are well documented, highly relevant and most of the conclusions well justified. However, the manuscript is not without flaws (see below) that should be addressed. I very much enjoyed reading the manuscript although parts of it are rather difficult and some of the arguments unnecessarily complicated and twisted.

Major points:

1. One of the claims repeatedly made in the ms, including the abstract, is that CD chromosome SC3 and in part chr.16 (SC10/20) affect vegetative growth (line154, line 179 and elsewhere). This is an over interpretation and misleading because conidiation is analyzed, not growth rate. In addition, conidiation is not carefully analyzed but inferred from simple OD measurements. And in the end, these findings are not essential for the main arguments. This must be either more carefully studied or left out of their arguments, particularly in the abstract. As it stands, it makes the interpretations less convincing.

Answer:

Thank you for supporting our manuscript. According to your suggestion, we changed statements about “vegetative growth” to “conidial formation” throughout the manuscript. In addition, in Abstract (L28), Introduction (L79) and Subtitle (L126), those statements were omitted. As you pointed out, conidial formation was analyzed only by measuring OD₆₀₀ value of conidial suspensions. To avoid concerns, we showed that the OD₆₀₀ value of conidial suspensions is linked to the amount of conidiospores (new Supplemental Figure 3).

2. The ms would be much easier to read if the authors had adopted the chromosome assignments of the recently published and highly similar genome of Fo5176, instead of referring to their own scaffolds of an incomplete assembly. This would also make Figure 2b and Figure 3a so much easier to understand.

Answer:

Thank you for your suggestions. We initially considered this but we have noticed some rearrangements (e.g. SC03, SC08; Figure 1b) and felt that it is difficult to correspond to the chromosome assignments of Fo5176. To make it easier to understand, we now showed as Chr^{SC10/20} and Chr^{SC16/18} in new Figures 2b and 3a and new Supplementary Figure 2b.

3. None of the pictures of affected plants is sufficiently detailed to be evaluated by the reader. The authors should provide larger pictures in the supplement.

Answer:

To help readers understand phenotypes and scoring, we now showed enlarged photos with disease index scores in new Supplementary Figure 4 and Supplementary Figure 12d. The original small pictures were removed from Figures.

Additional points:

Line 67: Maybe a rephrasing here could improve understanding the details.

For example by stating how SIX6 differs between the species.

Answer:

We changed to “*SIX1*, *SIX3*, *SIX5* and *SIX6* from *Fol* are involved in overcoming resistance in tomato and the *SIX6* homolog from *Forc016* is crucial for virulence in cucumber”.

(L66-67)

Line 93: maybe add a sentence to clarify what the situation is with the other SIX genes not mentioned here? Only the insiders would know how many there are.

Answer:

Thank you for your suggestion. We added a sentence “We did not detect homologs of any other *SIX* genes.”

(L95)

Line 98 and Supplemental data 1: The authors report on 265 secreted effectors, but in their Supplementary data 1 some appear to be identical e.g. FocnCong_v001238 and Focn_v001293. This should be clarified.

Answer:

There are indeed 265 effector candidate genes (265 loci). However, some genes encode exact same amino acid sequences. This is not the annotation error.

To assist readers, we now added the following sentence in footnotes of new Table 1 and new Supplementary Data 1.

“Note that some genes encode identical amino acid.”

Supplementary data 1: Please check spelling in footnotes; Why do you use the term “Target gene”?

Answer:

We checked and corrected spelling in footnotes. “Target gene sequence” and “Target gene translation” were changed to “Nucleotide sequence” and “Amino acid sequence”, respectively in new Supplementary Data 1.

Line 110: Please add quantitative data concerning TE content.

Answer:

According to your suggestion, quantitative data concerning TE content were shown in new Supplementary Figure 1.

We added the data in Results as follows;

“All known effector genes except *FOA4* are located in the TE-rich genomic region in *FocnCong:1-1* as follows: *SIX1* (in SC8), *SIX4* (SC9), *SIX8* (SC10), *SIX9* (SC3), *FOA1* (SC5), *FOA1b* (SC10), *FOA2* (SC9), *FOA3* (SC3), and *FOA4b* (SC10) (Supplementary Fig. 1).”

(L108-111)

Supplement data 2: Please check the formatting of commas and dots. In other words, why are the data not rounded? The authors should provide a complete list of expression data not just the effectors. Only then can expression of the selected genes here be compared to the expression of the core genome.

Answer:

According to your suggestion, the data were rounded off to the second decimal place, and the expression data for all genes were shown in new Supplementary Data 2. To make it easier to check only the expression of effector genes, the expression data for effector candidates were also added in new Supplementary Data 1.

Line 119: Here it seems at least to me that the fact that the genome reported here and that of isolate Fo5176 (reference 21) are >95 % identical might raise potential questions concerning the novelty of the genomic data. The authors should comment or at least they should discuss the differences between these two genomes more intensively here or elsewhere. For example how similar are the coding regions? Which genes are missing?

Answer:

Thank you for your comments. We noticed that there are differences in the two genomes and a detailed comparison between *FocnCong1-1* and Fo5176 would be very interesting if there are differences in especially pathogenicity. However, it is not the main content in this manuscript, and thus we prefer to describe it elsewhere.

Line 127: This is a critical experiment and a paragraph hard to read!

Benomyl treatment is obviously mutagenic, therefore the frequencies of chromosome loss should be pointed out here to judge how frequently these genotypes arise. The authors should also give some information as to whether or not additional rearrangements or mutations other than chromosome loss can be observed (The authors have the data to do this). In addition, the pulsed field gel of HS2 in figure 2 suggests that a genomic rearrangement has occurred here, not just a chromosome loss. Please comment.

The authors analysed only one strain for each karyotype. As a consequence, their conclusions regarding the function of particular CD chromosomes can be tricky. I understand that all science is the art of the doable, but I would be more careful here. Many additional mutations may occur that affect the phenotype that can easily be overlooked.

Maybe the authors can include more than just one mutant for the critical karyotype HS5. Maybe, cell sorting might do the job to generate these as here no mutagenic material is used. The authors should also provide a list of genes missing in the respective mutants HS1 to HS6. As they have the data, this should be easy.

Answer:

In this experiment, we did select hygromycin B-sensitive mutants after benomyl treatment using *FocnCong:1-1 ΔSIX4* in which *SIX4* had been replaced with a hygromycin B resistance gene cassette as a parental strain. Therefore, we do not know how frequently HS mutants arose (i.e. how many progenies were subjected for selection).

As you pointed out, benomyl treatment caused genome rearrangements. Genome sequencing of the HS mutants revealed that duplication of SC2 and part of SC2 and SC17 occurred in HS1, HS5 and HS2, respectively (new Figure 2a). We investigated phenotypes in an additional HS mutant with the same karyotype as HS5 (HS5L: HS5-like mutant). Like HS5, HS5L showed virulence on *cyp79b2/cyp79b3* and *pad3* plants, but not on Col-0 WT plants (new Supplementary Figure 12). We cannot rule out the possibility that these genome rearrangements and/or additional mutations affect phenotypes. We believe, however, that the return of HS5 virulence on Arabidopsis in two independent HS5 transformants containing *FocnCong1-1 SIX8-PSE1* (new Figure 5c) supports the main conclusion in this manuscript. We added and redrafted the sentences in Results and Discussion as follows;
“In addition, duplication of SC2 and part of SC2 and SC17 occurred in HS1, HS5 and HS2, respectively (Fig. 2a).”

(L137-138)

“In this study, *FocnCong:1-1* HSs were generated by treatment with the mitosis inhibitor benomyl. In the generation process, a genome rearrangement, but not just a chromosome loss, has occurred at least in HS1, HS2 and HS5 (Fig. 2a). We also investigated phenotypes in an additional HS mutant with the same karyotype as HS5 (HS5L: HS5-like mutant; Supplementary Fig. 12). Like HS5, HS5L showed virulence on *cyp79b2/cyp79b3* and *pad3* plants, but not on Col-0 WT plants. We cannot rule out the possibility that these genome rearrangements affect phenotypes. In addition to the results of HS5L, however, the return of HS5 virulence on Arabidopsis in two independent HS5 transformants containing *FocnCong1-1 SIX8-PSE1* (Fig. 5c) supports the conclusion that the *SIX8-PSE1* pair is required for virulence on Arabidopsis.”

(L333-342)

As the expression data for all genes were shown in new Supplementary Data 2, which genes are missing in the respective HS mutants can be found out.

Line 142 and Figure 2: line 975:

I am afraid that with such small pictures I cannot judge the course of infection. Full size pictures should be provided in the supplement.

Answer:

To help readers understand phenotypes and scoring, we now showed enlarged photos with disease index scores in new Supplementary Figure 4 and Supplementary Figure 12d. The original small pictures were removed from Figures.

Line 144: given that the assay used to measure conidiation is only based on OD measurements this may be an overstatement. Showing the plates (supplement) is fine but biomass could be measured more quantitatively. Conidiation would only be a concern if conidiation is required for virulence.

The effect on conidiation could be more intensively studied to support the strong statement in their abstract that the authors have identified an unusual CD chromosome affecting vegetative growth. Microscopy and counting the conidia may be helpful. The authors imply that loss of conidiation affects virulence, but the amount of inoculated bud cells is the same in all cases, so this, in my opinion, should not matter.

Line 151-254: The conclusion here is hard to follow:

There is no evidence that SC10/20 is associated with vegetative growth other than increased numbers of conidia.

Answer:

As you pointed out, conidial formation was analyzed only by measuring OD₆₀₀ value of conidial suspensions. To avoid concerns, we showed that the OD₆₀₀ value of conidial suspensions is linked to the amount of conidiospores (new Supplemental Figure 3).

According to your suggestion, we changed statements about “vegetative growth” to “conidial formation” throughout the manuscript. In addition, in Abstract (L28), Introduction (L79) and Subtitle (L126), those statements were omitted.

Line 157: Where was HS6 is tagged with phleomycin? If they know they should include this.

Answer:

We do not know where the phleomycin resistance gene (*ble*) is located in the HS6-BLE genome because the genome has not been sequenced.

Line 188:

In this paragraph it is not clear why the authors did not analyze strains HCT2, HCT3 and HCT4 because those would have been nice controls for the observed effects. This should be included. I also could not find any information about SIX4-GFP or about HS2-GFP. Maybe

include were to find it.

Answer:

On the basis of results before this paragraph using *FocnCong*:1-1 strains HSs and HCTs, we concluded that Chr^{SC10/SC20} is required for virulence on Arabidopsis. In this paragraph, we investigated if Chr^{SC10/SC20} is involved in suppression of *CYP79B2/CYP79B3*-mediated immunity. We believe that the current data using strains HSs support our conclusions. In addition, the results from an additional HS mutant (HS5L) described above also support the conclusions (new Supplementary Figure 12).

For Δ *SIX4*-GFP and HS2-GFP, we added the information to the Material and Method sections. (L451-454)

We sincerely apologize for omitting the information.

Line 221: the *pad3* mutant was more susceptible to HS5. The authors may in the future consider analysing *cyp71A27* as it is active in roots.

Answer:

Thank you for a great suggestion.

Line 229 the pair of effectors:

The transcription starts which can be deduced from their RNA Seq analysis should be included here, in order to get an idea about the distance of the promoters or the presence of a bidirectional promoter.

Answer:

According to your suggestion, on the basis of RNA seq results, the untranscriptional intergenic region was deduced and shown in new Supplementary Figure 6.

Line 242: complementation of HS5. While the data support the notion that both genes *SIX8* and *PSE1* are required for the biological effect, it cannot be excluded that in the strains harboring only *SIX8* or *PSE1* the lack of complementation is due to a lack of expression. So, it is still possible that only one of the two genes is required. Since this a strong statement in their title it might be a good idea to check expression here.

Answer:

Thank you for critical comments. According to your suggestion, expression levels of transgenes in *FocnCong*:1-1 HS5 transformants were checked (new Supplementary Figure 7c). For this purpose, we quantified expression levels of transgenes at 3 dpi in *cyp79b2/cyp79b3*. This is because expression levels of both *SIX8* and *PSE1* are low in bud cells and HS5:*SIX8* and HS5:*PSE1* cannot infect on Arabidopsis Col-0 WT (new Figure 5), but can infect on *cyp79b2/cyp79b3*. As shown in new Supplementary Figure 7c, we confirmed expression of transgenes. Note that, in the case of HS5:*PSE1* #2, *PSE1* expression

was very low, while it is fine in HS5:*PSE1* #1. We think that it was due to individual differences in infections.

Line 245: The effects of deleting the locus is an important experiment. Did it have to go to the supplement because the effect was so small?

Answer:

According to your suggestion, disease index results of Arabidopsis Col-0 WT challenged with *Focn*Cong:1-1 Δ *SIX8-PSE1* (previous Supplementary Figure 5c) were now shown in new Figure 5d. In addition, representative images with disease index scores were shown in new Supplementary Figure 4i.

Line 250: here the authors should provide more details concerning the conservation of *SIX8* and *PSE1* among Arabidopsis infecting oxysporum species, including either a sequence alignment or a quantitation (% identity). How for example does completely conserved differ from conserved?

Answer:

According to your suggestion, the details concerning the conservation of *SIX8* and *PSE1* among *F. oxysporum* isolates were provided as a quantitation (% identity) in new Figure 6a, b.

Line 250: Very nice to show that *SIX8-PSE1* locus is active from Arabidopsis infecting strain but not from a strain lacking virulence on Arabidopsis!

Answer:

Thank you for your supporting comment.

Line 323: The sequences should be compared here or elsewhere with regard to protein domains. Even if the protein sequences of *SIX8* and *PSE1* may not be particularly informative, the authors should say something about it, as the reader will be asking this.

Answer:

The protein sequences of *SIX8* and *PSE1* were subjected to motif searches, such as PROSITE and Pfam, and the results were no hit. We added the sentences in Discussion as follows; “As bioinformatic analysis of *SIX8* and *PSE1* protein sequences gives no known domain annotations, identification of host targets of *SIX8* and *PSE1* will be required to clarify functions of the paired effectors.”

(L360-363)

Line 342: Please rephrase.

Answer:

We redrafted the sentences in Discussion as follows;

“It is also notable that disruption or loss occurs in only *PSE1*, but not in *SIX8*, in certain non-Arabidopsis infecting *F. oxysporum* isolates. Perhaps *PSE1*, but not *SIX8*, is recognizable in plants that carry corresponding resistance proteins, leading to its disruption or loss to avoid detection.”

(L358-361)

Summary: Nice work.

Again, thank you very much for your helpful critical comments.

Response to the comments of Reviewer #2

This is an informative and well-reasoned manuscript describing the genomic features of a strain of *Fusarium oxysporum* that causes disease on cabbage and Arabidopsis. The authors sequence strain Cong 1-1 and compare it to genomes of other strains with the same or different host specializations. They identify features consistent with conditionally dispensable (CD) chromosomes and these as confirmed to be CD by mutagenesis studies. By testing various mutants they narrow down genomic scaffolds that, when missing, alter pathogenic phenotype. One such scaffold (SC10/SC20) is found to be required for virulence on Arabidopsis but not cabbage. Arabidopsis resistance is dependent upon the plant gene PAN3 involved in the synthesis of the phytoalexin camalexin thus the SC10/SC20 conferred virulence appears to suppress camalexin resistance in some manner. Two adjacent genes (SIX8 and PSE1) for predicted effectors found on SC10/SC20 when added together restore virulence when added back to a strain lacking the entire SC10/SC20. Neither gene alone restores virulence. Overall, this is an impressive piece of work describing the genomic features of this model phytopathogen and identifying specific genes and other regions of the genome of significance to pathogenicity. Other than described below, all major claims of the paper are justified. This will add to our knowledge of this important pathosystem.

My only comment and suggestion for improvement involves the section describing how fungal mutations affect infection phenotype (Figure 4b). First, the photos illustrating the phenotypes are very hard to see because they are so small. Can there be a supplementary figure showing the phenotypes in more detail and explaining exactly how the comparisons were conducted? This is not described in much detail in the Methods section. Because scoring of the phenotypes seem somewhat subject, were these phenotypic assessments conducted blind without knowledge of the genotype of the sample being examined? In the Figure, all comparisons were made to the deltaSIX4-GFP / Col-0 control which includes differences in both the plant and pathogen genotypes. But perhaps the more important comparison is the difference for the mutant HS5-GFP on Col-0 versus *cyp79b2/cyp79b3*. Can this be included?

Answer:

Thank you for supporting comments. According to your suggestion, we now showed enlarged photos with disease index scores in new Supplementary Figure 4 and Supplementary Figure 12d. The original small pictures were removed from Figures.

In addition, we added the result of statistical comparison between Col-0 and *cyp79b2/cyp79b3* inoculated with *Focn*Cong:1-1 HS5-GFP in new Figure 4b.

REVIEWERS' COMMENTS:

Reviewer #1 (Remarks to the Author):

Review Yu et al. 2020

The authors have revised their manuscript and carefully addressed all points I raised in my original review of the ms.

The authors have in the meantime found a technical flaw in one piece of their data as follows: First of all, we would like to explain about the changes that were not pointed out by Editor and Reviewers.

- - New Figure 5a:

We had used the primer set (FoTEF-Q2-F/R) reported by van der Does et al. (2008) to amplify Focn EF1 α as an internal control for qPCR. However, now we found that the primer set amplifies a band even from Arabidopsis-only sample. Therefore, we designed new primers (552 & 553) to amplify Focn β -tubulin (TUB2) gene as a control and reanalysed expression levels of SIX8 and PSE1 by qPCR, renewing the results of new Figure 5a. The primer sequences were shown in new Supplementary Table 3.

- - New Supplementary Figure 2b (previous Supplementary Figure 1b):

Since the previous electrophoretic photograph contained cropped data, the same experiment was performed and the new electrophoretic photograph replaced the old one.

I have carefully looked at these items and found the data did not alter any of the authors original claims.

In summary I think the authors did a very good job improving their nice and important paper.

Reviewer #2 (Remarks to the Author):

My concerns were minor for this manuscript, mostly involving disease phenotyping. The authors addressed these concerns so I am satisfied. No further revisions necessary.